# Gradient Multi-Normalization for Efficient LLM Training

**Meyer Scetbon**[*]
Microsoft Research

**Chao Ma**[*]
Microsoft Research

**Wenbo Gong**[*]
Microsoft Research

**Ted Meeds**
Microsoft Research

## Abstract

Training large language models (LLMs) commonly relies on adaptive optimizers such as Adam (Kingma & Ba, 2015), which accelerate convergence through moment estimates but incur substantial memory overhead. Recent stateless approaches such as SWAN (Ma et al., 2024) have shown that appropriate preprocessing of instantaneous gradient matrices can match the performance of adaptive methods without storing optimizer states. Building on this insight, we introduce *gradient multi-normalization*, a principled framework for designing stateless optimizers that normalize gradients with respect to multiple norms simultaneously. Whereas standard first-order methods can be viewed as gradient normalization under a single norm (Bernstein & Newhouse, 2024), our formulation generalizes this perspective to a multi-norm setting. We derive an efficient alternating scheme that enforces these normalization constraints and show that our procedure can produce, up to an arbitrary precision, a fixed-point of the problem. This unifies and extends prior stateless optimizers, showing that SWAN arises as a specific instance with particular norm choices. Leveraging this principle, we develop SinkGD, a lightweight matrix optimizer that retains the memory footprint of SGD (w/o momentum) while substantially reducing computation relative to whitening-based methods. On the memory-efficient LLaMA training benchmark (Zhao et al., 2024a), SinkGD achieves state-of-the-art performance, reaching the same evaluation perplexity as Adam using only 40% of the training tokens.

## 1   Introduction

The training of Large Language Models (LLMs) relies heavily on adaptive optimization algorithms, such as Adam Kingma & Ba (2015), which dynamically adjust learning rates for each parameter based on past gradient information, leading to faster convergence and improved stability. However, these optimizers introduce substantial memory overhead due to the storage of internal states, typically moment estimates of gradients, a challenge that becomes particularly pronounced in distributed training settings where memory constraints and communication overhead are critical concerns Rajbhandari et al. (2020); Korthikanti et al. (2023); Dubey et al. (2024). In contrast, simpler first-order optimization methods such as Stochastic Gradient Descent (SGD) require significantly less memory but fail to adequately train LLMs Zhao et al. (2024b); Zhang et al. (2020); Kunstner et al. (2023, 2024). As a result, there is an ongoing need for developing new optimization strategies that resolves the memory efficiency v.s. training performance dilemma for large-scale models training.

Recent research has made significant strides in improving memory efficiency of optimization. A key focus is on low rank compression of optimizer states (Hu et al., 2021; Lialin et al., 2023; Zhao et al., 2024a; Hao et al., 2024; Xu et al., 2024a; Zhang et al., 2024; Gong et al., 2025; Chen et al., 2024a; Zhu et al., 2024). More recently, a new approach that directly removes certain optimizer states started to emerge, and has shown stronger results compared with direct state compression (Ma

---

[*]Equal contribution. This work was done when Meyer Scetbon was affiliated with Microsoft Research.

39th Conference on Neural Information Processing Systems (NeurIPS 2025).

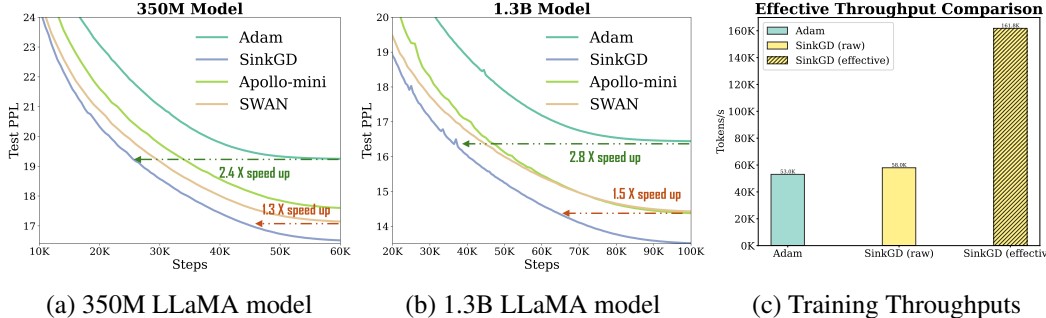

(a) 350M LLaMA model  (b) 1.3B LLaMA model  (c) Training Throughputs

Figure 1: `SinkGD` **performance preview on memory-efficient LLaMA pretraining benchmark Zhao et al. (2024a). (a)** and **(b)**: Comparison of the test perplexities obtained by Adam Kingma & Ba (2015); Zhao et al. (2024a), SWAN (Ma et al., 2024), Apollo (Zhu et al., 2024), and our proposed `SinkGD` (Algorithm 4) on 1B LLaMA pretraining task with C4 dataset. All loss curves of Adam and Apollo-mini are reproduced from the corresponding opensource codes. We also compare with their official results in Table 1. On both 350M and 1.3B LLama architectures, `SinkGD` achieves > 2× speed-up vs Adam in terms of tokens seen; and 1.3 to 1.5 X speed-up vs SWAN and Apollo. **(c)**: Training throughput analysis on training 1.3 B model on 8 × A100, under constant batch size = 130K tokens. We present two metrics: raw throughput, measured by number of training tokens consumed per second; and effective throughput, which is raw throughput adjusted by the token efficiency of optimizer relative to Adam. `SinkGD` has a raw throughput that is marginally higher than Adam, while improving the effective throughput by > 3×.

et al., 2024; Jordan et al., 2024; Chen et al., 2024c). Among these advancements, Ma et al. (2024) propose to replace all moving average states with a sequence of stateless matrix operations acting on the raw gradient tensors. This results in SWAN, an optimizer that achieved the same memory footprint as SGD (w/o momentum) while delivering comparable or even better performances than Adam. Collectively, they demonstrate that memory efficiency and loss throughput are not mutually exclusive. The goal of this paper is to understand the design principle of those empirical methods, and further improve their efficiency.

**Contributions.** Motivated by the insights of SWAN (Ma et al., 2024), we introduce *gradient multi-normalization*, a framework for designing efficient optimizers based on a novel multi-normalization scheme. Unlike standard first-order methods that can be interpreted as gradient normalization according to a single norm (Bernstein & Newhouse, 2024), our approach aims at normalizing gradients according to multiple norms. We demonstrate that gradient matrix processing-based techniques proposed in (Ma et al., 2024) can be understood as a particular instance of our approach with carefully chosen norms and shorter iterative steps. Given this new perspective, we instantiate a more efficient and scalable stateless optimizer (`SinkGD`) that achieves Adam-level computational cost, while having the same memory footprint as SGD. Moreover, it achieves on par or even outperforms Adam in LLM pretraining tasks, as well as various existing memory-efficient baselines under LLaMA architecture. Our contributions are summarized below:

- We propose a novel family of first-order methods, called *Multi-Normalized Gradient Descent* (MNGD), that aims at normalizing gradients according to multiple norms. Our framework generalizes the steepest descent viewpoint of Bernstein & Newhouse (2024) that recasts popular first-order optimizers as normalization of gradients under a single norm.

- We then propose a simple alternating scheme in order to effectively compute the multi-normalization of gradients, and show that our algorithm can provide a fixed-point solution up to an arbitrary precision, ensuring the normalization of its output with respect to the norms considered.

- We leverage our framework and design a new lightweight, stateless optimizer that improves the scalability of SWAN. Our algorithm, namely `SinkGD` (Algorithm 4), alternatively performs row-wise and column-wise normalization according to the Euclidean geometry. We show that `SinkGD` exactly recovers the square-root iterates of Sinkhorn algorithm (Sinkhorn, 1964).

- Finally, we evaluate our Sinkhorn-based stateless optimizer `SinkGD` by training LlaMA models on various scales, from 60m to 1.3B. Results show that `SinkGD` manages to be on par or even

outperforms the Adam optimizer, as well as other memory-efficient baselines. On the memory-efficient LLaMA training benchmark Zhao et al. (2024a), SinkGD achieves a 3× speedup over Adam with significantly reduced memory requirements.

## 2 Background

### 2.1 From Adam to Stateless Optimizers

**Adam Optimizer.** Adam (Kingma & Ba, 2015) relies on accumulating internal states throughout training in order to improve the convergence. More formally, given a loss function $(\theta, x) \in \Theta \times \mathcal{X} \to \mathcal{L}(\theta, x) \in \mathbb{R}$, where $\Theta \subset \mathbb{R}^d$ is the set of learnable parameters and $\mathcal{X}$ is the set where the data resides, Adam aims at minimizing $\theta \to \mathbb{E}_{x \sim \mathbb{P}_x}(\mathcal{L}(\theta, x))$ where $\mathbb{P}_x$ is the distribution of data on $\mathcal{X}$. To achieve this, Adam computes at every step $t \geq 1$ a stochastic gradient associated with a mini-batch of input data $x^{(t)}$, and performs the following updates (denoting $\nabla_t = \nabla_\theta \mathcal{L}(\theta_t, x^{(t)})$):

$$\mathrm{m}_t = \beta_1 \mathrm{m}_{t-1} + (1 - \beta_1)\nabla_t, \quad \mathrm{s}_t = \beta_2 \mathrm{s}_{t-1} + (1 - \beta_2)\nabla_t^{\odot 2},$$

$$\hat{\mathrm{m}}_t = \frac{\mathrm{m}_t}{1 - \beta_1^t}, \quad \hat{\mathrm{s}}_t = \frac{\mathrm{s}_t}{1 - \beta_2^t}, \quad \theta_{t+1} = \theta_t - \eta_t \frac{\hat{\mathrm{m}}_t}{\sqrt{\hat{\mathrm{s}}_t} + \varepsilon}$$

where $\odot$ is the Hadamard product, $\eta_t > 0$ are global step-sizes, and $\beta_1, \beta_2 > 0$ are the weights of the exponential moving averages (EMAs) for the first and second moments respectively. During training, Adam optimizer stores two additional states $(\mathrm{m}_t, \mathrm{s}_t)$, effectively tripling the memory required to train the model compared to a simple stochastic gradient descent (SGD) scheme.

**SWAN: a Stateless Optimizer.** Ma et al. (2024) propose to move away from the paradigm of keeping track of internal states, and propose SWAN, a stateless optimizer that only pre-processes the stochastic gradients before updating the parameters. More precisely, they propose to update the learnable weight matrices involved in the model using two matrix operators. Given a weight matrix $W \in \mathbb{R}^{m \times n}$, with $m \leq n$, at time $t \geq 1$, the SWAN update is (denoting $\nabla_t = \nabla_W \mathcal{L}(W_t, x^{(t)})$,):

$$\tilde{\nabla}_t = \sqrt{n}Q(\nabla_t)^{-1}\nabla_t, \quad \hat{\nabla}_t = \sqrt{n}(\tilde{\nabla}_t\tilde{\nabla}_t^\top)^{-1/2}\tilde{\nabla}_t, \quad W_{t+1} = W_t - \eta_t\hat{\nabla}_t \tag{1}$$

where for a matrix $W \in \mathbb{R}^{m \times n}$, $Q(W) := \mathrm{Diag}(\|W_{1,:}\|_2, \ldots, \|W_{m,:}\|_2)$ is the diagonal matrix of size $m$ where the diagonal coefficients are the $\ell_2$-norm of the rows of $W$. To compute $(\hat{\nabla}_t\hat{\nabla}_t^\top)^{-1/2}$, the authors leverage the Newton-Schulz algorithm (Song et al., 2022; Li et al., 2018; Huang et al., 2019) instead of computing the SVD. While this approach does not require storing any additional states, it still suffers from a computational burden due to the $\mathcal{O}(m^2(n + m))$ computation of $(\nabla_t\nabla_t^\top)^{-1/2}\nabla_t$ which may limit its practical usage for certain scenarios.

### 2.2 Steepest Descent as Gradient Normalization

Bernstein & Newhouse (2024) interpret several gradient descent schemes as steepest descent methods under specific norms. More formally, they propose to minimize a local quadratic model of the loss $\mathcal{L}(\cdot, x^{(t)})$ at $\theta_t$ w.r.t to a given norm $\|\cdot\|$, that is:

$$\mathcal{Q}_{\|\cdot\|}(z) := \mathcal{L}(\theta_t, x^{(t)}) + \langle\nabla_t, z\rangle + \frac{\lambda_t}{2}\|z\|^2$$

where $\lambda_t > 0$ are the sharpness parameters and $\nabla_t := \nabla_\theta\mathcal{L}(\theta_t, x^{(t)})$ is the current stochastic gradient. As shown in (Bernstein & Newhouse, 2024), finding a minimizer of $\mathcal{Q}_{\|\cdot\|}$ can be equivalently formulated as solving:

$$-\frac{\|\nabla_t\|_*}{\lambda_t} \argmax_{z \in \mathbb{R}^d:\ \|z\|=1} \langle\nabla_t, z\rangle \tag{2}$$

where $\|x\|_* := \sup_{z \in \mathbb{R}^d:\ \|z\|=1}\langle x, z\rangle$ is the dual norm of $\|x\|$. Several popular gradient-descent schemes can be recovered using the above approach. For example, when the $\ell_2$-norm is used, one recovers standard gradient descent, while the $\ell_\infty$ leads to signed gradient descent (Carlson et al., 2015). However, this framework considers only a single norm for pre-processing the raw gradient $\nabla_t$. In the following, we extend this approach to incorporate multiple norms for gradient pre-processing, enabling the design of efficient and stateless optimizers for LLM training.

# 3 Multi-Normalized Gradient Descent

**Notations.** For a vector $x \in \mathbb{R}^d$, we call its normalized projection w.r.t to a given norm $\|\cdot\|$, the solution to the following optimization problem:

$$\mathcal{P}_{\|\cdot\|}(x) := \arg\max_{z:\ \|z\|=1}\langle x, z\rangle \tag{3}$$

We also extend the definition of this notation if $x \in \mathbb{R}^{m \times n}$ is a matrix and $\|\cdot\|$ is a matrix norm.

## 3.1 Gradient Multi-Normalization

Let us now consider a finite family of $K \geq 1$ norms $(g_1, \ldots, g_K)$. In order to pre-process the gradient $\nabla$ jointly according to these norms, we propose to consider the following optimization problem:

$$\arg\max_z \langle \nabla, z\rangle \text{ s.t. } \forall\, i \in [|1, K|],\ g_i(z) = 1\ . \tag{4}$$

Assuming the constraint set is non-empty, the existence of a maximum is guaranteed. However, this problem is NP-hard and non-convex due to the constraints, making it hard to solve efficiently for the general case of arbitrary norms.

---

**Algorithm 1** `MultiNorm`$(\nabla, L, \boldsymbol{g})$

---

**Input:** the stochastic gradient $\nabla_\theta \mathcal{L}(\theta_t, x^{(t)})$, the norms $\boldsymbol{g} := (g_1, \ldots, g_K)$, and $L \geq 1$ the number of iterations.
Initialize $x = \nabla_\theta \mathcal{L}(\theta_t, x^{(t)})$.
**for** $\ell = 1$ **to** $L$ **do**
  **for** $i = 1$ **to** $K$ **do**
    $x \leftarrow \mathcal{P}_{g_i}(x) := \arg\max_{z:\ g_i(z)=1} \langle x, z\rangle$
  **end for**
**end for**
Return $x$

---

**Algorithm 2** Multi-Normalized GD (`MNGD`)

---

**Input:** $T \geq 1$ the number of updates, $(\eta_t)_{0 \leq t \leq T}$ the global step-sizes, $\mathcal{L}$ the loss to minimize, $L \geq 1$ the number of iterations for the multi-normalization, and $\boldsymbol{g} := (g_1, \ldots, g_K)$ the norms.
**for** $t = 1$ **to** $T$ **do**
  $\nabla_t \leftarrow \nabla_\theta \mathcal{L}(\theta_t, x^{(t)})$ with $x^{(t)} \sim P_x$
  $\hat{\nabla}_t \leftarrow$ `MultiNorm`$(\nabla_t, L, \boldsymbol{g})$ as in Alg. 1.
  $\theta_{t+1} \leftarrow \theta_t - \eta_t \hat{\nabla}_t$
**end for**
Return $x$

---

*Remark* 3.1. Observe that when $K = 1$, the problem (4) recovers exactly the single normalization step used in Bernstein & Newhouse (2024).

*Remark* 3.2. The convex relaxation of (4), defined as

$$\arg\max_z \langle \nabla, z\rangle \quad \text{s.t. } \forall\ i \in [|1, K|],\ g_i(z) \leq 1 \tag{5}$$

is in fact equivalent to the single normalization case discussed in Section 2.2, where the norm considered is $\|x\| := \max_{i \in [|1, K|]} g_i(x)$. Thus, solving (5) is equivalent to computing the projection $\mathcal{P}_{\|\cdot\|}(\nabla)$. In Appendix C, we provide a general approach to compute it using the so-called Chambolle-Pock algorithm Chambolle & Pock (2011).

While solving (4) exactly might not be practically feasible in general, we propose a simple alternating projection scheme, presented in Algorithm 1. Notably, our method assumes that the projections $\mathcal{P}_{g_i}(\cdot)$ can be efficiently computed for all $i \in [|1, K|]$. Fortunately, when the $g_i$'s correspond to $\ell_p$-norms with $p \in [|1, +\infty|]$, or Schatten $p$-norms for matrices, closed-form solutions for these projections exist. See Appendix C for more details.

**SWAN: an Instance of** `MultiNorm`. SWAN Ma et al. (2024) applies two specific pre-processing steps to the raw gradients in order to update the weight matrices. In fact, each of these pre-processing steps can be seen as normalized projections with respect to a specific norm. More precisely, for $W \in \mathbb{R}^{m \times n}$ and $m \leq n$, let us define

$$g_1(W) := \frac{\max_{i \in [|1, m|]} \|W_{i,:}\|_2}{\sqrt{n}}\ , \text{ and } g_2(W) := \frac{\|W\|_{\sigma, \infty}}{\sqrt{n}}\ .$$

where for $p \in [1, +\infty]$, $\|W\|_{\sigma, p}$ is the Schatten $p$-norm of $W$. Simple derivations leads to the following equalities:

$$\mathcal{P}_{g_1}(W) = \sqrt{n} Q(W)^{-1} W, \quad \mathcal{P}_{g_2}(W) = \sqrt{n}(WW^\top)^{-1/2} W$$

Therefore applying a single iteration ($L = 1$) of Algorithm 1 with norms $g_1$ and $g_2$ as defined above on the raw gradient $\nabla_t$ exactly leads to the SWAN update (Eq. (1)).

### 3.2 On the Convergence of `MultiNorm`

We aim now at providing some theoretical guarantees on the convergence of `MultiNorm` (Algorithm 1). More precisely, following the SWAN implementation Ma et al. (2024), we focus on the specific case where $K = 2$ and the normalized projections associated with the norms $g_1$ and $g_2$ have constant $\ell_2$-norm. More formally, we consider the following assumption.

**Assumption 3.3.** Let $g$ be a norm on $\mathbb{R}^d$. We say that it satisfies the assumption if for all $x \in \mathbb{R}^d$, $\|\mathcal{P}_g(x)\|_2 = c$ where $c > 0$ is an arbitrary positive constant independent of $x$ and $\|\cdot\|_2$ represents the Euclidean norm.

*Remark* 3.4. Observe that both norms in SWAN satisfies Assumption 3.3 and their normalized projections have the same $\ell_2$-norm, as for any $W \in \mathbb{R}^{m \times n}$ with $m \leq n$, we have $\|\mathcal{P}_{g_1}(W)\|_2 = \|\mathcal{P}_{g_2}(W)\|_2 = \sqrt{nm}$.

This assumption enables to obtain useful properties on $\mathcal{P}_g$ as we show in the following Lemma:

**Lemma 3.5.** *Let $g$ a norm satisfying Assumption 3.3. Then*

$$\mathcal{P}_g \circ \mathcal{P}_g = \mathcal{P}_g$$

*and for all $x \in \mathbb{R}^d$, $g^*(\mathcal{P}_g(x)) = \|\mathcal{P}_g(x)\|_2^2 = c^2$, where $g^*$ is the dual norm associated with $g$.*

Let us now introduce some additional notation to clearly state our result. Let $x_0 \in \mathbb{R}^d$ and let us define for $n \geq 0$:

$$x_{2n+1} := \mathcal{P}_{g_1}(x_{2n}), \quad x_{2n+2} := \mathcal{P}_{g_2}(x_{2n+1}) \tag{6}$$

which is exactly the sequence generated by Algorithm 1 when $K = 2$ and $x_0 = \nabla_\theta \mathcal{L}(\theta_t, x^{(t)})$. Let us now show our main theoretical result, presented in the following Theorem.

**Theorem 3.6.** *Let $g_1$ and $g_2$ two norms on $\mathbb{R}^d$ satisfying Assumption 3.3 and such that their normalized projections have the same $\ell_2$ norm. Let also $(x_n)_{n \geq 0}$ be defined as in (6) and let us define the set of fixed-point as:*

$$\mathcal{F} := \{x : \mathcal{P}_{g_1}(x) = \mathcal{P}_{g_2}(x) = x\}$$

*Then by denoting $d(x, \mathcal{F}) := \min_{z \in \mathcal{F}} \|x - z\|_2$ we have*

$$d(x_n, \mathcal{F}) \xrightarrow[n \to \infty]{} 0 .$$

This Theorem states that if `MultiNorm` runs for a sufficient amount of time, then the returned point $x$ can be arbitrarily close to a fixed-point solution. While we cannot guarantee that it solves (4), we can assert that our algorithm converges to a fixed-point solution with arbitrary precision, and as a by-product produces a solution $x$ normalized w.r.t both norms $g_1, g_2$ (up to an arbitrary precision).

*Remark* 3.7. Note that in Theorem 3.6 we assume that the normalized projections associated to $g_1$ and $g_2$ have the same $\ell_2$-norms. However, given two norms $g_1$ and $g_2$ satisfying Assumption 3.3, i.e. such that for all $x$:

$$\|\mathcal{P}_{g_1}(x)\|_2 = c_1, \quad \|\mathcal{P}_{g_2}(x)\|_2 = c_2$$

for some $c_1, c_2 > 0$, and given a target value $a > 0$, one can always rescale the norms such that their normalized projections have the same $\ell_2$ norm equal to $a$. More formally, by denoting $\tilde{g}_1 = \frac{c_1}{a} g_1$ and $\tilde{g}_2 = \frac{c_2}{a} g_2$, we obtain that

$$\|\mathcal{P}_{\tilde{g}_1}(x)\|_2 = \|\mathcal{P}_{\tilde{g}_2}(x)\|_2 = a.$$

*Remark* 3.8. It is worth noting that, for squared matrices ($m = n$), a single iteration ($L = 1$) of `MultiNorm` using the norms considered in Ma et al. (2024), immediately converges to a fixed-point—precisely recovering SWAN.

### 3.3 MNGD: a New Family of Stateless Optimizers.

We now introduce our family of optimizers: *Multi-Normalized Gradient Descents* (MNGDs) (Algorithm 2). The key distinction from the framework proposed in Bernstein & Newhouse (2024) is that MNGDs normalize the gradient with respect to multiple norms using the `MultiNorm` step, whereas in Bernstein & Newhouse (2024), the gradient is normalized using a single norm. In the following, we focus on the MNGD scheme with a specific choice of norms, for which we can efficiently compute the gradient multi-normalization step. This enables the application of stateless optimizers to LLMs.

## 4 Sinkhorn: a Multi-Normalization Procedure

As in SWAN Ma et al. (2024), we propose to normalize the weight matrices according to multiple norms. We still leverage the row-wise $\ell_2$-norm to pre-process raw gradients, however, rather than using the spectral norm, we propose to consider instead a relaxed form of this constraint and use the column-wise $\ell_2$-norm. More formally, consider the two following norms on matrices of size $\mathbb{R}^{m \times n}$:

$$g_1(W) := \frac{\max_{i \in [|1,m|]} \|W_{i,:}\|_2}{\sqrt{n}} \ , \quad g_2(W) := \frac{\max_{j \in [|1,n|]} \|W_{:,j}\|_2}{\sqrt{m}} \ ,$$

which leads to the following two normalized projections:

$$\mathcal{P}_{g_1}(W) = \sqrt{n}Q(W)^{-1}W, \quad \mathcal{P}_{g_2}(W) = \sqrt{m}WR(W)^{-1}$$

where $R(W) := \text{Diag}(\|W_{:,1}\|_2, \ldots, \|W_{:,n}\|_2) \in \mathbb{R}^{n \times n}$ is the diagonal matrix of size $n$ with the $\ell_2$-norm of the columns of $W$ as diagonal coefficients. For such a choice of norms, the `MultiNorm` reduces to a simple procedure as presented in Algorithm 3.

*Remark* 4.1. For such a choice of norms, we obtain $\|\mathcal{P}_{g_1}(W)\|_2 = \|\mathcal{P}_{g_2}(W)\|_2 = \sqrt{nm}$ for any $W \in \mathbb{R}^{m \times n}$. That is, both norms satisfy Assumption 3.3 and their $\ell_2$ norms are equal to $\sqrt{nm}$.

For completeness we include the MNGD scheme (Algorithm 4) that replaces the `MultiNorm` step with `SR-Sinkhorn` (Algorithm 3).

---

**Algorithm 3** `SR-Sinkhorn`$(\nabla, L)$

**Input:** the stochastic gradient $\nabla_W \mathcal{L}(W_t, x^{(t)})$, and $L \geq 1$ the number of iterations.
Initialize $X = \nabla_W \mathcal{L}(W_t, x^{(t)}) \in \mathbb{R}^{m \times n}$.
**for** $\ell = 1$ **to** $L$ **do**
  $Q(X) = \text{Diag}(\|X_{1,:}\|_2, \ldots, \|X_{m,:}\|_2)$
  $X \leftarrow \sqrt{n}Q(X)^{-1}X$
  $R(X) = \text{Diag}(\|X_{:,1}\|_2, \ldots, \|X_{:,n}\|_2)$
  $X \leftarrow \sqrt{m}XR(X)^{-1}$
**end for**
Return $X$

---

**Algorithm 4** Sinkhorn GD (`SinkGD`)

**Input:** $T \geq 1$ the number of updates, $(\eta_t)_{0 \leq t \leq T}$ the global step-sizes, $\mathcal{L}$ the loss to minimize, and $L \geq 1$ the number of iterations for the SR-Sinkhorn procedure.
**for** $t = 1$ **to** $T$ **do**
  $\nabla_t \leftarrow \nabla_\theta \mathcal{L}(\theta_t, x^{(t)})$ with $x^{(t)} \sim P_x$
  $\hat{\nabla}_t \leftarrow$ `SR-Sinkhorn`$(\nabla_t, L)$ as in Alg. 3.
  $\theta_{t+1} \leftarrow \theta_t - \eta_t \hat{\nabla}_t$
**end for**
Return $x$

---

**The Sinkhorn Algorithm.** Before explicitly showing the link between Algorithm 3 and the Sinkhorn algorithm, let us first recall the Sinkhorn theorem Sinkhorn (1964) and the Sinkhorn algorithm Sinkhorn & Knopp (1967). Given a positive coordinate-wise matrix $A \in \mathbb{R}_+^{m \times n}$, there exists a unique matrix $P \in \mathbb{R}_+^{m \times n}$ of the form $P = QAR$ with $Q$ and $R$ positive coordinate-wise and diagonal matrices of size $m$ and $n$ respectively, such that $P\mathbf{1}_n = n\mathbf{1}_m$ and $P^\top\mathbf{1}_m = m\mathbf{1}_n$. To find $P$, one can use the Sinkhorn algorithm that initializes $P_0 := A$ and computes for $k \geq 0$:

$$P_{k+1/2} = n\text{Diag}(P_k\mathbf{1}_n)^{-1}P_k, \quad P_{k+1} = mP_{k+1/2}\text{Diag}(P_{k+1/2}^\top\mathbf{1}_m)^{-1} \ .$$

Equivalently, these updates on $P$ can be directly expressed as updates on the diagonal coefficients of $Q = \text{Diag}(u)$ and $R = \text{Diag}(v)$ with $u \in \mathbb{R}_+^m$ and $v \in \mathbb{R}_+^n$. By initializing $u_0 = \mathbf{1}_m$ an $v_0 = \mathbf{1}_m$, the above updates can be reformulated as follows:

$$u_{k+1} = n\frac{\mathbf{1}_m}{Av_k}, \quad v_{k+1} = m\frac{\mathbf{1}_n}{A^\top u_{k+1}} \tag{7}$$

where $/$ denote the coordinate-wise division. Franklin & Lorenz (1989) show the linear convergence of Sinkhorn's iterations. More formally, they show that $(u_k, v_k)$ converges to some $(u^*, v^*)$ such that $P := \mathrm{Diag}(u^*)A\mathrm{Diag}(v^*)$ satisfies $P\mathbf{1}_n = n\mathbf{1}_m$ and $P^\top \mathbf{1}_m = m\mathbf{1}_n$, and:

$$d_{\mathcal{H}}(u_k, u^*) \in \mathcal{O}(\lambda(A)^{2k}) , \text{ and } d_{\mathcal{H}}(v_k, v^*) \in \mathcal{O}(\lambda(A)^{2k}) ,$$

where $d_{\mathcal{H}}$ is the Hilbert projective metric De La Harpe (1993) and $\lambda(A) < 1$ is a contraction factor associated with the matrix $A$.

**Links between Sinkhorn and Algorithm 3.** Algorithm 3 can be seen as a simple reparameterization of the updates presented in (7). More precisely, given a gradient $\nabla \in \mathbb{R}^{m \times n}$ and denoting $A := \nabla^{\odot 2}$, we obtain that the iterations of Algorithm 3 exactly compute:

$$u_{k+1}^{1/2} = \sqrt{n \frac{\mathbf{1}_m}{Av_k}}, \ \ v_{k+1}^{1/2} = \sqrt{m \frac{\mathbf{1}_n}{A^\top u_{k+1}}} \tag{8}$$

where the square-root is applied coordinate-wise, and returns after $L$ iterations $X_L = \mathrm{Diag}(u_L^{1/2})\nabla\mathrm{Diag}(v_L^{1/2})$. Therefore the linear convergence of Algorithm 3 follows directly from the convergence rate of Sinkhorn, and Algorithm 3 can be thought as applying the square-root Sinkhorn algorithm, thus the name `SR-Sinhkorn`. Note also that at convergence ($L \to +\infty$) we obtain $X^* \in \mathbb{R}^{m \times n}$ which is a fixed-point of both normalized projections, that is $\mathcal{P}_{g_1}(X^*) = \mathcal{P}_{g_2}(X^*) = X^*$, from which we deduce that $\|X_{i,:}^*\|_2 = \sqrt{n}$ and $\|X_{:,j}^*\|_2 = \sqrt{m}$ as demonstrated in Theorem 3.6.

**On the Importance of the Scaling.** Now that we have shown the convergence `SR-Sinkhorn`, let us explain in more detail the scaling considered for both the row-wise and column-wise normalizations. First recall that both norm $g_1$ and $g_2$ satisfy Assumption 3.3 and that the $\ell_2$ norm of their normalized projections is equal to $\sqrt{nm}$. The reason for this specific choice of scaling ($\sqrt{nm}$) is due to the global step-size in Algorithm 4. In our proposed MNGD, we did not prescribe how to select $\eta_t$. In practice, we aim to leverage the same global step-sizes as those used in Adam( Kingma & Ba (2015)) for training LLMs, and therefore we need to globally rescale the (pre-processed) gradient accordingly. To achieve that, observe that when EMAs are turned-off, Adam corresponds to a simple signed gradient descent, and therefore the Frobenius norm of the pre-processed gradient is simply $\sqrt{nm}$. Thus, when normalizing either the rows or the columns, we only need to rescale the normalized gradient accordingly.

**Computational Efficiency of SinkGD over SWAN.** Compared to SWAN Ma et al. (2024), the proposed approach, `SinkGD`, is more efficient as it only requires $\mathcal{O}(nm)$ numerical operations. In contrast, SWAN, even when implemented with Newton-Schulz, still requires performing matrix-matrix multiplications, which have a time complexity of $\mathcal{O}(m^2(m+n))$. In the next section, we will demonstrate the practical effectiveness of MNGD with `SR-Sinkhorn`, that is `SinkGD`. This approach manages to be on par with, and even outperforms, memory-efficient baselines for pretraining the family of LLaMA models up to 1B scale.

# 5 Experimental Results

In this section, we evaluate the empirical performance of applying `SinkGD` optimizer to LLM pretraining tasks. All experiments were performed on NVIDIA A100 GPUs.

## 5.1 LlaMA Pre-training Tasks

**Setup.** We evaluate **SinkGD** on the memory-efficient LLaMA training benchmark proposed by Zhao et al. (2024a). This benchmark uses LLaMA-based architecture (Touvron et al., 2023) with RMSNorm and SwiGLU activations (Zhang & Sennrich, 2019; Gao et al., 2023). We consider models with 60M, 130M, 350M, and 1.3B parameters, all trained on the C4 dataset Raffel et al. (2020) using an effective token batch size of 130K tokens (total batch size 512, context length 256). Specifically, for both 130M and 350M, we use 128 batch size with 4 accumulations. For 60M and 1B, we uses 256 batch with 2 accumulation, and 32 per-device batch size with 2 accumulation and 8xA100s, respectively. Following the setup of Zhao et al. (2024a); Zhu et al. (2024), **SinkGD** is applied to all linear modules in both attention and MLP blocks with $L = 5$ iterations for the `SR-Sinkhorn` procedure. For all other modules, that are the embedding layer, the RMSnorm layers, and the last

Table 1: Comparison with Adam and memory-efficient baselines on pre-training LLaMA models with C4 dataset. Test PPL is reported, along with a memory estimate of the total of parameters and optimizer states in BF16 format. The PPLs reported for all competitive methods are taken from Zhao et al. (2024a); Zhu et al. (2024) and Ma et al. (2024). We report both the reported Adam results from Zhao et al. (2024a) and our reproduced result. Note that the memory estimations from Zhao et al. (2024a); Zhu et al. (2024) did not consider the fact that Adam optimizer was used for embedding layers. This is corrected in our estimates.

| Methods | 60M | | 130M | | 350M | | 1.3B | |
|---|---|---|---|---|---|---|---|---|
| | PPL | MEM | PPL | MEM | PPL | MEM | PPL | MEM |
| Adam (reproduced) | 33.94 | 0.32G | 25.03 | 0.75G | 19.24 | 2.05G | 16.44 | 7.48G |
| Adam (cited) | 34.06 | 0.32G | 25.08 | 0.75G | 18.80 | 2.05G | 15.56 | 7.48G |
| Galore | 34.88 | 0.26G | 25.36 | 0.57G | 18.95 | 1.29G | 15.64 | 4.43G |
| Fira | 31.06 | 0.26G | **22.73** | 0.57G | 17.03 | 1.29G | 14.31 | 4.43G |
| Apollo-mini | 31.93 | 0.23G | 23.53 | 0.43G | 17.18 | 0.93G | 14.17 | 2.98G |
| Apollo | 31.55 | 0.26G | 22.94 | 0.57G | 16.85 | 1.29G | 14.20 | 4.43G |
| SWAN | 32.28 | 0.23G | 24.13 | 0.43G | 18.22 | 0.93G | 15.13 | 2.98G |
| SinkGD | **30.99** | 0.23G | 22.75 | 0.43G | **16.51** | 0.93G | **13.51** | 2.98G |
| SinkGD speed up v.s. Adam (reproduced) | 1.60 X | | 1.56 X | | 2.42 X | | 2.79 X | |
| SinkGD speed up v.s. Adam (cited) | 1.66 X | | 1.73 X | | 2.10 X | | 2.17 X | |
| Total Training Steps | 10K | | 20K | | 60K | | 100K | |

output layer, **Adam** optimizer Kingma & Ba (2015) is used. We use exactly the same cosine learning rate scheduler as in Zhao et al. (2024a), where $10\%$ of total training steps is used for warm-up. Note that, as in Zhao et al. (2024a); Zhu et al. (2024), we use a group-wise learning rate for our optimizer. The effective learning rate used for linear modules in the transformer blocks is of the form $\alpha\eta_t$ where $\eta_t$ is global learning rate provided by the scheduler and $\alpha$ is fixed hyperparameter that we set to $\alpha = 0.05$. For Adam, we use $\eta_t$ as the learning rate.

**Baselines.** We consider the following memory-efficient optimizers baselines: **Adam** (Kingma & Ba, 2015); **Galore** Zhao et al. (2024a); **Fira** Chen et al. (2024b), **Apollo** and **Apollo-mini** Zhu et al. (2024), and **SWAN** Ma et al. (2024). For all methods, training uses BF16 precision for weights, gradients and optimizer states by default, except for **SWAN** that uses FP32 precision to pre-process the gradient Ma et al. (2024). We also perform a grid search of learning rate for Adam over $\{0.01, 0.005, 0.001, 0.0005, 0.0001\}$, except for 1B model which we search over $\{0.001, 0.0007, 0.0005, 0.0003, 0.0001\}$. We do not perform any weight decay for all optimizers.

Table 2: Comparison of the test perplexities obtained during training when training 1B LLaMA with SinkGD v.s. 7B LLaMA using different baselines. For Apollo, Apollo-mini, 8-bit Adam and Galore, we cite the number from Zhu et al. (2024).

| Method | Mem. | 40K | 80K | 120K | 150K |
|---|---|---|---|---|---|
| 8-bit Adam (7B) | 26G | 18.09 | 15.47 | 14.83 | 14.61 |
| 8-bit Galore (7B) | 18G | 17.94 | 15.39 | 14.95 | 14.65 |
| Apollo (7B) | 15.03G | 17.55 | 14.39 | 13.23 | 13.02 |
| Apollo-mini (7B) | 13.53G | 18.03 | 14.60 | 13.32 | 13.09 |
| SinkGD (1B) | 2.98G | **16.44** | **14.27** | **13.17** | **12.97** |

**Performance evaluation and memory efficiency analysis.** The results presented in Table 1 demonstrate the effectiveness of **SinkGD** in terms of both computational efficiency and model performance. Notably, **SinkGD** achieves competitive performance while maintaining the lowest estimated memory consumption, comparable to that of SGD. Across all evaluated models, our method performs on par with or even surpasses **Adam** and other memory-efficient baselines in terms of test perplexity. In particular, **SinkGD** outperforms all other baselines in this experimental setup for the 350M and 1.3B model variants. Additionally, we quantify the computational efficiency of **SinkGD** by measuring the speed-up relative to **Adam**. This is determined by computing the ratio of the total training steps of **Adam** to the number of steps needed for **SinkGD** to reach the same final test perplexity. Note also that the reported memory consumption values in Table 1 account for three components: (1) memory allocated for model parameters, (2) optimizer-related costs for linear modules within transformer blocks, and (3) Adam's memory footprint for remaining parameters.

Table 3: Raw and effective throughput analysis.

| Method | Raw / eff. throughput |
|--------|----------------------|
| Adam | 53047 / 53047 (tokens/s) |
| SinkGD | 57982 / 161769 (tokens/s) |

**Comparative analysis of 1B and 7B LLaMA training.** To further evaluate the efficacy of our proposed optimizer, we replicate the experimental setup of Zhu et al. (2024), but instead train a 1B-parameter LLaMA model using **SinkGD** and compare its performance against a 7B-parameter LLaMA model trained with **Apollo**, **Apollo-mini**, **8-bit Adam**, and **8-bit Galore**. As shown in Table 2, the 1B model trained with **SinkGD** achieves comparable test perplexities to those of the 7B model trained with **Apollo** after 150K optimization steps, while incurring significantly lower costs. Notably, training the 7B LLaMA model with Apollo requires **15** days on an 8xA100 GPU setup to complete 150K steps, whereas our approach achieves a similar loss in **3.3** days. The reported memory estimates correspond to the total memory cost detailed in the previous paragraph.

## 5.2 Ablation Study

**Throughput analysis.** We also assess throughput when training a 1.3B-parameter model on 8xA100 GPUs. We use two metrics: (1) the *raw throughput* which is the number of tokens processed per second, and (2) the *effective throughput* defined as the total training token used by Adam divided by the time (in seconds) used by **SinkGD** to reach the same test perplexities. These metrics evaluate the impact of the multi-normalization step on training speed, and also account for the fact that some optimizers make more effective use of training tokens. As shown in Table 3, **SinkGD** achieves competitive raw throughput compared to **Adam**, suggesting the multi-normalization step does not require expensive computations. Furthermore, **SinkGD** exhibits a $3 \times$ higher effective throughput than Adam, indicating a significantly faster wall-clock time convergence.

**On the effect of the number of iterations.** In this experiment, we measure the effect of applying different iterations of our proposed `MultiNorm` (Algorithm 1) scheme in the specific case of the **SWAN** and **SinkGD** methods. More specifically, we train a 130M LLaMA model on C4 datasets and compare the test perplexities obtained after 10K steps. We observe a small but consistent improvement when using $L = 5$ iterations, we decide to use this number of iterations in our benchmark evaluation, as reported in table 1.

Table 4: Comparison of the test PPLs obtained during training at 10K steps when training 130M LLaMA model with either SWAN or SinkGD using different number of iterations in `MultiNorm`.

| Method | PPL |
|--------|-----|
| SWAN ($L = 1$) | 26.79 |
| SWAN ($L = 5$) | 26.56 |
| SinkGD ($L = 1$) | 26.21 |
| SinkGD ($L = 5$) | 26.13 |

## 6 Related Work

**Gradient Normalization.** Gradient normalization has emerged as a key technique in optimization, complementing its well-established role in forward-pass operations such as Layer Normalization (LayerNorm) (Ba et al., 2016). LARS and LAMB (You et al., 2017, 2019) employ global normalization to raw gradients and Adam's layer-wise updates, respectively, improving convergence and mitigating gradient pathologies in large-batch training. Apollo (Zhu et al., 2024) introduces a channel-wise scaling approach, while SWAN (Ma et al., 2024) replaces Adam's first-moment estimate with normalized gradients to stabilize gradient distributions. Theoretical analyses further underscore the importance of gradient normalization. Hazan et al. (2015) study its convergence properties in SGD, while Cutkosky & Mehta (2020) demonstrate that incorporating momentum enhances convergence without requiring large batches. Bernstein & Newhouse (2024) interpret normalization in certain optimizers as a form of steepest descent under a specific norm, with SignSGD (Bernstein et al., 2018), or standard gradient descent, serving as examples of gradient normalization. More recently, the concurrent work of Vyas et al. considered iterative processing of gradient matrices from the perspective of whitening.

**Memory Efficient Optimizers.** Optimizers for large-scale training can reduce memory consumption primarily through two approaches: (1) low-rank approximation and (2) elimination of internal state dependencies. Low-rank optimizers project gradients onto a reduced subspace, allowing internal state updates within this subspace. ReLoRA (Lialin et al., 2023) periodically merges LoRA (Hu et al., 2021) weights to restore full-rank representations. FLoRA (Hao et al., 2024) employs random Gaussian projections, whereas GaLore (Zhao et al., 2024a) utilizes singular value decomposition (SVD) for structured projections, further improved by Fira (Chen et al., 2024a) via a compensation term. Apollo (Zhu et al., 2024) minimizes memory overhead using rank-1 state representations. An alternative approach eliminates the need for internal states altogether. SWAN (Ma et al., 2024) removes Adam's first and second moments through gradient normalization and whitening. Adam-mini (Zhang et al., 2024) reduces memory by leveraging block-wise second moment estimation. SGD-SaI (Xu et al., 2024b) obviates Adam's second moment by precomputing learning rate scaling. Sign-based optimization (Chen et al., 2024c) enables large-scale training using only first-moment updates. Muon (Jordan et al., 2024), a simplification of Shampoo (Gupta et al., 2018), accelerates large model training via whitened first-moment updates, further demonstrating the viability of reduced-memory optimizers. (Gong et al., 2025) unified memory-efficient optimizers under the perspective of structured Fisher information approximation, and proposed low rank approximation with error corrections to reduce memory and accelerate computation.

**Alternating Projection.** Many iterative fixed-point algorithms employ alternating updates to enforce constraints or refine estimates. A classical example is the Von Neumann algorithm Von Neumann (1950), which alternates projections onto affine subspaces and converges to their intersection. The Sinkhorn algorithm Sinkhorn & Knopp (1967) similarly alternates row and column normalizations, which can be seen as Bregman projections Benamou et al. (2015) onto affine spaces, to approximate entropy-regularized optimal transport. While effective in Hilbert spaces, these algorithms do not generalize to arbitrary convex sets. Dykstra's algorithm Dykstra (1983) extends these methods by introducing correction terms, ensuring convergence to the exact projection. More generally, alternating projection methods have been extended through Pierra's product space reformulation Pierra (1984), as well as modern techniques like ADMM Boyd et al. (2011) and block-coordinate methods Tibshirani (2017) in large-scale optimization. Despite these theoretical advances, extending alternating projection methods to non-convex settings remains a significant challenge. Recent progress includes manifold-based projection methods Lewis & Malick (2008), and proximal alternating techniques Bolte et al. (2014), which aim to improve convergence in non-convex problems, yet a comprehensive theory for convergence remains an open question.

## 7  Conclusion.

In this work, we introduced *gradient multi-normalization*, a novel optimizer design framework. Our approach formalizes gradient matrix processing as the normalization of stochastic gradients with respect to multiple norms, and we propose an alternating optimization procedure to achieve this normalization efficiently. We establish that our multi-normalization scheme can approximate, to arbitrary precision, a fixed point of the optimization problem, thereby ensuring that the gradient is appropriately scaled according to both norms. Using this principle, we developed SinkGD, a lightweight matrix optimizer with reduced computational cost while retaining the same memory footprint as SGD. Experiments on pretraining LLaMA models with up to 1B parameters demonstrate a strong speedup over Adam with significantly reduced memory requirements. Open questions for future research include: 1, understanding the mathematical mechanisms of how gradient multi-normalization enables the removal of EMA states, and when such approaches will fail; 2, the understanding of the theoretical properties of multi-normalization-based optimization; and 3, extending the applicability of gradient multi-normalization to other training tasks beyond LLM pre-training, as well as its scalability to frontier model size training regimes.

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

# Appendix

## A  Implementation details

**General setup**  We describe the implementation setups for the LLM pre-training tasks. To enable a more straightforward and comparable analysis, we simply replicate the setting of Zhao et al. (2024a), under exactly the same model configs and optimizer hyperparameter configs, whenever possible. This includes the same model architecture, tokenizer, batch size, context length, learning rate scheduler, learning rates, subspace scaling, etc.

**Precision**  All baselines uses BF16 for model weights, gradients, and optimizer states storage. For SWAN and SWAN$^\dagger$, we follow the original paper and use FP32 in there whitening step.

**Learning rate scheduling**  we use exactly the same scheduler as in Zhao et al. (2024a) for all methods.

**Hyperparameters**  Since **SinkGD** utilizes matrix-level operations on gradients, it can only be applied to 2D parameters. Therefore, in our experiments, we only apply **SinkGD** on all linear projection weights in transformer blocks. Similar to Galore (Zhao et al., 2024a), the rest of the non-linear parameters still uses Adam as the default choice. Therefore, we follow the learning rate setup of Galore, where we fix some global learning rate across all model sizes and all modules. Then, for the linear projection modules where **SinkGD** is applied, we simply apply a scaling factor $\alpha$ on top of the global learning rate. For all **SinkGD**, we adopt a *lazy-tuning approach* (hyperparameters are set without extensive search), as detailed below. This helps to reduce the possibility of unfair performance distortion due to excessive tuning.

- **Adam**  For Adam we use same learning rate tuning procedure as suggested by Zhao et al. (2024a) and Ma et al. (2024) (i.e., performing grid search over $\{0.01, 0.005, 0.001, 0.0005, 0.0001\}$). We found that the optimal learning rates for Adam is 0.001. The only exception is that for a model of size 1.3B: as we already know that a larger model requires smaller learning rates, we conduct a learning search for Adam over a smaller but more fine-grained grid of $\{0.001, 0.0007, 0.0005, 0.0003, 0.0001\}$. As a result, the optimal learning rate found for Adam on 1.3B is 0.0007. Other hyperparameters of Adam ($\beta_1$, $\beta_2$, $\epsilon$ etc) follows the implementation of (Zhao et al., 2024a), that is, we use $\beta_1 = 0.9$, $\beta_2 = 0.999$. We also perform further ablation on extensive grid-search over all hyperparameters, see Appendix D.1.

- **SWAN**$^\dagger$, is the tuned version of SWAN presented in Ma et al. (2024). The original results of **SWAN** from Ma et al. (2024) assumes no learning rate warm-up and no learning rate tuning, in order to demonstrate the robustness of the method. This setting is more challenging than the setting of the usual literature (Zhao et al., 2024a; Zhu et al., 2024). Hence, for fair comparison we relax those constraints and matches the setting of Galore and Apollo: we now allow learning rate warm-up (set to the same as Adam and Apollo), as well as larger learning rates for SWAN. This improved version of SWAN is denoted by **SWAN**$^\dagger$. We use a global learning rate of 0.02, as well as the scaling factor $\alpha = 0.05$. This is selected by simply searching the learning rate over a constraint grid $\{0.01, 0.02, 0.05\}$, and then setting $\alpha = 0.05$ such that the effective learning rate is scaled back to 0.001. Finally, for other hyperparameters, we follow Ma et al. (2024).

- Finally, **SinkGD**, we use the same global learning rate of 0.02, as well as the scaling factor $\alpha = 0.05$ which are the same as **SWAN**$^\dagger$, across all model sizes. We suspect with more careful tuning, its performance can be significantly improved; however, this is out of the scope of the paper. For SR-Sinkhorn$(\nabla, L)$ operation used in **SinkGD**, we simply use 5 steps.

# B Proofs

## B.1 Proof of Lemma 3.5

*Proof.* Let us assume that $\|\mathcal{P}_g(x)\|_2 = c$ and so for any $x$. Then we have that:

$$\|\|\mathcal{P}_g \circ \mathcal{P}_g(x)\|_2 \|\mathcal{P}_g(x)\|_2 \geq g^*(\mathcal{P}_g(x)) := \langle \mathcal{P}_g \circ \mathcal{P}_g(x), \mathcal{P}_g(x) \rangle$$
$$\geq \sup_{z:\, g(z) \leq 1} \langle z, \mathcal{P}_g(x) \rangle$$

where the first inequality follows from Cauchy–Schwarz and the second inequality follows from the definition of $\mathcal{P}_g$. Now recall by definition, that $g(\mathcal{P}_g(x)) \leq 1$, and therefore we can select $z = \mathcal{P}_g(x)$ in the right inequality which gives:

$$\|\mathcal{P}_g \circ \mathcal{P}_g(x)\|_2 \|\mathcal{P}_g(x)\|_2 \geq g^*(\mathcal{P}_g(x))$$
$$\geq \|\mathcal{P}_g(x)\|_2^2$$

However because $\|\mathcal{P}_g \circ \mathcal{P}_g(x)\|_2 = \|\mathcal{P}_g(x)\|_2 = c$, we obtain that

$$g^*(\mathcal{P}_g(x)) = \|\mathcal{P}_g(x)\|_2^2$$

and by optimality, we also deduce that $\mathcal{P}_g \circ \mathcal{P}_g(x) = \mathcal{P}_g(x)$. $\qquad\square$

## B.2 Proof of Thoerem 3.6

*Proof.* First observe that thanks to Lemma 3.5, we have for any $n \geq 1$:

$$\|x_n\|_2^2 = g_1^*(x_{2n-1}) = g_2^*(x_{2n}) = c^2 \tag{9}$$

where $g_1^*$ and $g_2^*$ are the dual norms of $g_1$ and $g_2$ respectively. We also have that for $n \geq 1$

$$g_2(x_{2n}) \leq 1, \; g_1(x_{2n+1}) \leq 1 \tag{10}$$

by definition of the normalized projections. We even have

$$g_2(x_{2n}) = g_1(x_{2n+1}) = 1$$

by optimality of the normalized projections. Let assume now that $n \geq 2$ is even, then we have that:

$$\langle x_{n+1}, x_n \rangle = \langle \mathcal{P}_{g_1}(x_n), x_n \rangle = g_1^*(x_n)$$
$$\geq \langle z, x_n \rangle \; \forall \, z \in \mathcal{B}_1(0_d)$$

where $\mathcal{B}_{g_1}(0_d)$ is the unit ball centered in $0_d$ associated to the norm $g_1$ and the inequality follows from the definition of $\mathcal{P}_{g_1}$. In particular by taking $z = x_{n-1} = \mathcal{P}_{g_1}(x_{n-2}) \in \mathcal{B}_{g_1}(0_d)$, we obtain that:

$$\langle x_{n+1}, x_n \rangle \geq \langle x_{n-1}, x_n \rangle$$

A similar proof can be conducted when $n$ is odd using the definition of $\mathcal{P}_{g_2}$. Therefore the sequence $(\langle x_{n+1}, x_n \rangle)_{n \geq 1}$ is increasing and bounded so it converges to a certain constant $r > 0$. From this result we directly deduces that:

- $(g_1^*(x_{2n}))_{n \geq 1}$ is monotonic increasing and converges towards $r$.

- $(g_2^*(x_{2n+1}))_{n \geq 1}$ is monotonic increasing and converges towards $r$.

Because $(x_{2n+1})_{n \geq 0}$ and $(x_{2n})_{n \geq 0}$ are bounded, we can extract a common subsequence $(x_{2\phi(n)+1})_{n \geq 1}$ and $(x_{2\phi(n)})_{n \geq 1}$ that converge to some cluster points $x_1$ and $x_2$ respectively.

Now by continuity of the dual norms and of the inner product we obtain that:

$$\lim_{n \to \infty} g_2^*(x_{2\phi(n)+1}) = g_2^*(x_1)$$
$$\lim_{n \to \infty} g_1^*(x_{2\phi(n)}) = g_1^*(x_2)$$
$$\lim_{n \to \infty} \langle x_{2\phi(n)+1}, x_{\phi(n)} \rangle = \langle x_1, x_2 \rangle$$

However observe that these three sequences are subsequences of $(\langle x_n, x_{n+1}\rangle)_{n\geq 0}$ which converges towards $r$, therefore we obtain that:

$$r = g_2^*(x_1) = g_1^*(x_2) = \langle x_1, x_2\rangle$$

Additionally, remark that

$$g_2^*(x_{2\phi(n)+1}) = g_2^*(\mathcal{P}_{g_1}(x_{2\phi(n)})) \tag{11}$$

Let us now show that $x_{2\phi(n)+1} = \mathcal{P}_{g_1}(x_{2\phi(n)}) \xrightarrow[n\to\infty]{} \mathcal{P}_{g_1}(x_2)$. Indeed we have that:

$$\langle \mathcal{P}_{g_1}(x_{2\phi(n)}), x_{2\phi(n)}\rangle = g_1^*(x_{2\phi(n)}) \xrightarrow[n\to\infty]{} g_1^*(x_2)$$

Then, because $(\mathcal{P}_{g_1}(x_{2\phi(n)}))_{n\geq 0}$ is bounded, we can extract a subsequence that converges towards $z$ such that $g_1(z) \leq 1$, from which follows that:

$$\langle z, x_2\rangle = \langle \mathcal{P}_{g_1}(x_2), x_2\rangle$$

then by optimality of $\mathcal{P}_{g_1}(x_2)$ over the unit ball induced by $g_1$, we deduce that $z = \mathcal{P}_{g_1}(x_2)$. This is true for all converging sub-sequences of $(\mathcal{P}_{g_1}(x_{2\phi(n)}))_{n\geq 0}$, therefore we have that $\mathcal{P}_{g_1}(x_{2\phi(n)}) \xrightarrow[n\to\infty]{} \mathcal{P}_{g_1}(x_2)$, and by unicity of the limit, it follows that

$$x_1 = \mathcal{P}_{g_1}(x_2) \ .$$

Now from the equality $g_2^*(x_1) = \langle x_1, x_2\rangle$, and given the fact that $g_2(x_2) \leq 1$ (as for all $n$ $g_2(x_{2\phi(n)}) \leq 1$ which is obtained from (10)), we deduce that

$$x_2 = \mathcal{P}_{g_2}(x_1)$$

thanks to the optimality of $\mathcal{P}_{g_2}$. Now observe now that:

$$g_2^*(x_1) = g_2^*(\mathcal{P}_{g_1}(x_2)) = \langle \mathcal{P}_{g_2} \circ \mathcal{P}_{g_1}(x_2), \mathcal{P}_{g_1}(x_2)\rangle$$
$$= \langle x_2, \mathcal{P}_{g_2}(x_2)\rangle$$

where the equality follows from the fact that:

$$\mathcal{P}_{g_2} \circ \mathcal{P}_{g_1}(x_2) = \mathcal{P}_{g_2}(x_1) = x_2$$

and the two equalities follows the previous results obtained. Therefore we obtain that

$$g_2^*(x_1) = g_2^*(x_2) = c^2$$

where the last equality follows from (9). Thus, we obtain that

$$r = c^2 = g_2^*(x_1) = \langle x_1, x_2\rangle \leq \|x_1\|_2\|x_2\|_2$$

but from (9), $\|x_1\|_2 = \|x_2\|_2 = c$, from which follows that $x_1 = x_2 = x$, and $\mathcal{P}_{g_1}(x) = \mathcal{P}_{g_2}(x) = x$. As a by-product, we also obtain that $\langle x_n, x_{n+1}\rangle \xrightarrow[n\to\infty]{} r = c^2$, and therefore $\|x_n - x_{n-1}\|_2^2 = 2c^2 - 2\langle x_n, x_{n+1}\rangle \xrightarrow[n\to\infty]{} 0$.

From the above proof, we also conclude that if $y$ is a cluster point of $(x_n)_{n\geq 0}$, then there exists $\psi$ such that $(x_{\psi(n)})_{n\geq 0}$ converges towards $y$ that satisfies: $\mathcal{P}_{g_1}(y) = \mathcal{P}_{g_2}(y) = y$. Indeed this follows simply from the fact that we can extract a subsequence of $(x_{\psi(n)})_{n\geq 0}$ which has all indices that are either even or odd.

Let us now show that

$$g_1(x_n) \xrightarrow[n\to\infty]{} 1, \text{ and } g_2(x_n) \xrightarrow[n\to\infty]{} 1 \ .$$

Indeed for a convergent subsequence, if the subsequence has infinitely many odd indices the result is trivial from the fact $g_1(x_{2n+1}) = 1$. Now if the indices are even, we obtain that $g_1(x_{2\phi(n)}) \xrightarrow[n\to\infty]{} g_1(x)$, however $x$ has to be a fixed-point so $g_1(x) = g_1(\mathcal{P}_{g_1}(x)) = 1$. This hold for any subsequences, therefore we have $g_1(x_n) \xrightarrow[n\to\infty]{} 1$. Similarly, we can apply the same reasoning for $g_2(x_n)$.

Let us now show the following Lemma.

**Lemma B.1.** *Let $g_1$ and $g_2$ two norms satisfying the same assumption as in Theorem 3.6, that is for all $x$, $\|\mathcal{P}_{g_1}(x)\|_2 = \|\mathcal{P}_{g_2}(x)\|_2 = c$ with $c > 0$. Then by denoting $\mathcal{S}_g$ the unit sphere associated to a norm $g$, we have:*

$$\mathcal{S}_{g_1} \cap \mathcal{S}_{g_2} \cap \mathcal{S}_{c\ell_2} = \mathcal{F} \ \text{ where } \mathcal{F} := \{x : \ \mathcal{P}_{g_1}(x) = \mathcal{P}_{g_2}(x) = x\} \ .$$

*Proof.* Indeed $\mathcal{F} \subset \mathcal{S}_{g_1} \cap \mathcal{S}_{g_2} \cap \mathcal{S}_{c\ell_2}$ follows directly from the definition of $\mathcal{P}_{g_1}$, $\mathcal{P}_{g_1}$, and from Assumption 3.3. Now let $z \in \mathcal{S}_{g_1} \cap \mathcal{S}_{g_2} \cap \mathcal{S}_{c\ell_2}$. Observe that

$$c^2 \geq \langle z, \mathcal{P}_1(z)\rangle = \sup_{q:\ g_1(q)=1} \langle z, q\rangle$$

where the inequality follows from the assumption on $\mathcal{P}_{g_1}$ and from the definition of $z$. Then as $g_1(z) = 1$, we deduce that:

$$c^2 \geq \langle z, \mathcal{P}_1(z)\rangle \geq \|z\|_2^2 = c^2$$

from which folows that $\mathcal{P}_{g_1}(z) = z$. Similarly we deduce that $\mathcal{P}_{g_2}(z) = z$, and thus we have $\mathcal{S}_{g_1} \cap \mathcal{S}_{g_2} \cap \mathcal{S}_{c\ell_2} \subset \mathcal{F}$. $\qquad\square$

Now observe that $d(x_n, \mathcal{S}_{g_1}) \xrightarrow[n\to\infty]{} 0$, and $d(x_n, \mathcal{S}_{g_1}) \xrightarrow[n\to\infty]{} 0$. Additionally, from (11), we have $d(x_n, \mathcal{S}_{c\ell_2}) = 0$, therefore we have that $d(x, \mathcal{S}_{g_1} \cap \mathcal{S}_{g_2} \cap \mathcal{S}_{c\ell_2}) \xrightarrow[n\to\infty]{} 0$ since all these spaces are closed, and the result follows from Lemma B.1.

$\qquad\square$

## C   On the Convex Relaxation of Problem (4)

Given $K$ norms, $(g_1, \ldots, g_K)$, in this section we are interested in solving:

$$\arg\max_z \langle \nabla, z\rangle \quad \text{s.t. } \forall\ i \in [\![1, K]\!], \ g_i(z) \leq 1 \tag{12}$$

which as stated in the main paper is equivalent to solve

$$\arg\max_z \langle \nabla, z\rangle \quad \text{s.t. } \|z\| \leq 1$$

where

$$\|z\| := \max_{i \in [\![1, K]\!]} g_i(z) \ . \tag{13}$$

Note that this constrained optimization problem is exactly finding the subdifferential of the dual norm $\|\cdot\|$. To see this, let us recall the following Lemma with its proof Watson (1992).

**Lemma C.1.** *The subdifferential of a norm $\|\cdot\|$ at $x$ is given by*

$$\partial\|\cdot\|(x) = \{p \in \mathbb{R}^d : \ \|p\|_* \leq 1, \ \langle p, x\rangle = \|x\|\}$$

*where the dual norm is defined as*

$$\|x\|_* := \max_{\|z\| \leq 1} \langle z, x\rangle$$

*Proof.* We can show this result by double inclusion. Let us define the subdifferential of a norm as

$$\partial\|\cdot\|(x) := \{p : \ \|y\| \geq \|x\| + \langle p, y - x\rangle \ \forall y\}$$

and let us denote our set of interest as

$$\mathcal{V}(x) := \{p \in \mathbb{R}^d : \ \|p\|_* \leq 1, \ \langle p, x\rangle = \|x\|\}$$

Let $p \in \mathcal{V}(x)$. Then we have

$$\begin{aligned}
\|x\| + \langle p, y - x\rangle &= \langle p, y\rangle \\
&\leq \|p\|_* \|y\| \\
&\leq \|y\|
\end{aligned}$$

where the first equality comes from the definition of $p$, the first inequality comes from Holder, and the last one is obtained by definition of $p$. So we deduce that $\mathcal{V}(x) \subset \partial \|\cdot\|(x)$. Let us now take $p \in \partial \|\cdot\|(x)$, that is $p$ such that for all $y$ $\|y\| \geq \|x\| + \langle p, y - x \rangle$. Then we have for all $y$ that:

$$\langle p, x \rangle - \|x\| \geq \langle p, y \rangle - \|y\|$$
$$\geq \sup_y \langle p, y \rangle - \|y\|$$
$$\geq \|p\|^*$$

where $\|\cdot\|^*$ is the Fenchel-Legendre transform of the norm $\|\cdot\|$. From Lemma C.3, we deduce that

$$\langle p, x \rangle - \|x\| \geq \mathbf{1}_{\mathcal{B}_1}(p)$$

where $\mathcal{B}_1$ is the unit ball associated with the dual norm $\|\cdot\|_*$. As the left-hand side is finite, we deduce that $p \in \mathcal{B}_1$. Then we deduce that

$$\|x\| \geq \langle p, x \rangle \geq \|x\|$$

where the left inequality is obtained by applying Holder to the inner-product, from which follows the result.

$\square$

For simple norms, such as the $\ell_p$-norm with $1 < p \leq +\infty$, obtaining an element of $\partial \|\cdot\|_*(\nabla)$ can be done in closed-form. For $\ell_p$ norms, recall the their dual norm are the $\ell_q$ norms with $q$ the dual exponent respectively. The following Lemma provide analytic formulas for such norms.

**Lemma C.2.** *For $x \in \mathbb{R}^d$, let us define the $\ell_p$-norm as*

$$\|x\|_p := \left( \sum_{i=1}^d |x_i|^p \right)^{1/p}$$

***Case 1: $1 < p < \infty$.** Let us define the dual exponent $q$ by*

$$\frac{1}{p} + \frac{1}{q} = 1.$$

*Then the subdifferential of $\|x\|_p$ is*

$$\partial \|x\|_p = \begin{cases} \left\{ \dfrac{(|x_1|^{p-2}x_1, \ldots, |x_d|^{p-2}x_d)}{\|x\|_p^{p-1}} \right\}, & x \neq 0, \\ \{ g \in \mathbb{R}^d : \|g\|_q \leq 1 \}, & x = 0. \end{cases}$$

***Case 2: $p = 1$.** For the $\ell_1$-norm, the subdifferential at $x \in \mathbb{R}^n$ is given by*

$$\partial \|x\|_1 = \left\{ g \in \mathbb{R}^n : g_i = \text{sign}(x_i) \right\}.$$

*Here, $\text{sign}(x_i)$ is $+1$ if $x_i > 0$, $-1$ if $x_i < 0$, and can be any value in $[-1, 1]$ if $x_i = 0$.*

***Case 3: $p = \infty$.** For the $\ell^\infty$-norm, let $M = \|x\|_\infty$ and let us define*

$$\mathcal{S}(x) := \{ g : g_i = 0 \text{ if } |x_i| < M, \; g_i = \text{sign}(x_i) \text{ else, } \|g\|_1 = 1 \}$$

*Then by denoting for any set $A \subset \mathbb{R}^d$, conv($A$) the convex hull of the set $A$, we have:*

$$\partial \|x\|_\infty = \begin{cases} \text{conv}(\mathcal{S}(x)) & x \neq 0, \\ \{ g \in \mathbb{R}^n : \|g\|_1 \leq 1 \}, & x = 0. \end{cases}$$

However, for general norms, there are not known closed-form solutions of their associated subdifferentials. In particular, if the norm is defined as in (13), even when the $g_i$'s are simple norms (i.e. norms for which we can compute the subdifferential of their dual norms), then no closed-form solution can be obtained in general.

## C.1 A Dual Perspective

In this section, we propose an algorithmic approach to solve the convex relaxation of the problem introduced in (4). More formally, given a family of simple norms $(g_i)_{i=1}^K$ and some positive constants $(\varepsilon_i)_{i=1}^K$, we consider the following problem:

$$\max_{d\theta \in \mathbb{R}^d} \langle \nabla, d\theta \rangle \quad \text{s.t. } g(d\theta) \leq 1 . \tag{14}$$

where

$$g(x) := \max_{i \in [[1,K]]} \frac{g_i(x)}{\varepsilon_i}$$

which is also a norm. For such problems, as long as $\nabla \neq 0$, then the solutions lies in the level set $\{d\theta : g(d\theta) = 1\}$. Even if the subdifferentials of (the dual norm of) each $g_i$ can be derived in closed form, there is not known closed-form for the subdifferential of (the dual norm of) $g$. To solve (14), we propose to consider a coordinate gradient descent on the dual. A simple application of the Fenchel duality Rockafellar (1974) leads to the following equivalent optimization problem:

$$\inf_{\lambda_1, \dots, \lambda_K} \sum_{i=1}^K \epsilon_i g_i^\dagger(\lambda_i) \quad \text{s.t.} \quad \nabla \mathcal{L}(\theta) = \sum_{i=1}^K \lambda_i \tag{15}$$

where $g_i^\dagger$ is the dual norm of $g_i$ and so for all $i \in [[1, K]]$, from which a primal solution can be recovered by simply finding $y_i$ s.t. $\lambda_i > 0$ and such that $\langle \lambda_i, y_i \rangle = \varepsilon_i g_i^\dagger(y_i)$ under the condition that $g_i(y_i) = \varepsilon_i$, which is equivalent to solve:

$$y_i^* := \varepsilon_i \arg\max_{z:\, g_i(z) \leq 1} \langle z, \lambda_i \rangle .$$

*Proof.* Let $(\mathcal{B}_i(\epsilon_i))_{i=1}^K$ the ball associated with the norm $(g_i)_{i=1}^K$ with radius $(\varepsilon_i)_{i=1}^K$ respectively. Let us also denote for any set $\mathcal{A} \subset \mathbb{R}^d$, the indicator function as

$$\mathbf{1}_A(x) = \begin{cases} 0 & \text{if } x \in A \\ +\infty & \text{otherwise} \end{cases}$$

In the following we denote $f(x) := \langle x, \nabla \rangle$. Then (14) can be reformulated as the following optimization problem:

$$-\inf_{d\theta} f(d\theta) + \sum_{i=1}^K \mathbf{1}_{\mathcal{B}_i(\epsilon_i)}(d\theta)$$

which can be again reparameterized (up to the sign) as

$$\inf_{x=y_i,\ \forall i \in [[1,K]]} f(x) + \sum_{i=1}^K \mathbf{1}_{\mathcal{B}_i(\epsilon_i)}(y_i)$$

Now the Lagrangian associated with this problem is:

$$\mathcal{F}((\lambda_i)_{i=1}^K, (y_i)_{i=1}^K, x) :=$$

$$f(x) - \langle x, \sum_{i=1}^K \lambda_i \rangle + \sum_{i=1}^K \mathbf{1}_{\mathcal{B}_i(\epsilon_i)}(y_i) + \langle y_i, \lambda_i \rangle$$

And taking the infimum of the Lagrangian w.r.t the primal variables leads to the following optimization problem:

$$\inf_x f(x) - \langle x, \sum_{i=1}^K \lambda_i \rangle + \sum_{i=1}^K \inf_{y_i} \mathbf{1}_{\mathcal{B}_i(\epsilon_i)}(y_i) + \langle y_i, \lambda_i \rangle$$

Now observe that

$$\inf_x f(x) - \langle x, \sum_{i=1}^{K} \lambda_i \rangle = -\sup_x \langle x, \sum_{i=1}^{K} \lambda_i \rangle - f(x)$$

$$= -f^*(\sum_{i=1}^{K} \lambda_i)$$

where $f^*$ is the Fenchel-Legendre transform of $f$. Similarly, we have:

$$\inf_{y_i} \mathbf{1}_{\mathcal{B}_i(\epsilon_i)}(y_i) + \langle y_i, \lambda_i \rangle = -\sup_{y_i} \langle y_i, -\lambda_i \rangle - \mathbf{1}_{\mathcal{B}_i(\epsilon_i)}(y_i)$$

$$= -\mathbf{1}^*_{\mathcal{B}_i(\epsilon_i)}(-\lambda_i)$$

Finally the dual of the problem is:

$$\sup_{\lambda_1,...,\lambda_K} -f^*(\sum_{i=1}^{K} \lambda_i) - \sum_{i=1}^{K} \mathbf{1}^*_{\mathcal{B}_i(\epsilon_i)}(-\lambda_i)$$

Now recall that $f(x) := \langle x, \nabla \rangle$, therefore we have that

$$f^*(x) = \mathbf{1}_{\{\nabla\}}(x)$$

Also, we have that

$$\mathbf{1}^*_{\mathcal{B}_i(\epsilon_i)}(x) = \varepsilon_i g_i^\dagger(x)$$

where $g_i^\dagger$ is the dual norm of $g_i$, from which it follows the final dual formulation:

$$\inf_{\lambda_1,...,\lambda_K} \sum_{i=1}^{K} \epsilon_i g_i^\dagger(\lambda_i) \quad \text{s.t.} \quad \nabla \mathcal{L}(\theta) = \sum_{i=1}^{K} \lambda_i.$$

Finally, Slater condition are verified, thus strong duality holds, and the KKT conditions gives the following primal-dual conditions:

$$\begin{cases} \nabla \mathcal{L}(\theta) = \sum_{i=1}^{K} \lambda_i \\ \lambda_i \in \partial \mathbf{1}_{\mathcal{B}_i(\varepsilon_i)}(y_i) \ \forall i \\ x = y_i \ \forall i \end{cases}$$

Now according to Lemma C.4, we have that

$$\partial \mathbf{1}_{\mathcal{B}_i(\varepsilon_i)}(x) = \begin{cases} \{0\} \text{ if } g_i(x) < \varepsilon_i \\ \emptyset \text{ if } g_i(x) > \varepsilon_i \\ \{p : \ \langle p, x \rangle = \varepsilon_i g_i^\dagger(p)\} \text{ if } g_i(x) = \varepsilon_i \end{cases}$$

from which follows that one can recover a primal solution by simply finding $y_i$ s.t. $\lambda_i > 0$ and such that $\langle \lambda_i, y_i \rangle = \varepsilon_i g_i^\dagger(y_i)$ under the condition that $g_i(y_i) = \varepsilon_i$, which is equivalent to solve:

$$y_i^* := \varepsilon_i \arg\max_{z: \ g_i(z) \leq 1} \langle z, \lambda_i \rangle .$$

$\square$

To solve the dual problem introduced in (15), we apply a coordinate gradient descent on the $\lambda_i$. More precisely, we can reformulate the problem as an unconstrained optimization one by considering:

$$\inf_{\lambda_2,...,\lambda_K} \epsilon_1 g_1^\dagger \left( \nabla \mathcal{L}(\theta) - \sum_{i=2}^{K} \lambda_i \right) + \sum_{i=2}^{K} \epsilon_i g_i^\dagger(\lambda_i)$$

---

**Algorithm 5** Primal-Dual Algorithm to solve (16)

---

**Input:** $\beta_k \in \mathbb{R}^d$, $\epsilon_1, \epsilon_k > 0$, $\eta_1, \eta_2 > 0$, s.t. $\eta_1 \eta_2 < 1$.
Initialize $\lambda = z = u = 0_d$.
**for** $i = 1$ **to** $L$ **do**
    $\lambda_{\text{old}} \leftarrow \lambda$
    $z \leftarrow \text{proj}_{\mathcal{B}_1(\epsilon_1)}(z + \eta_1(u - \beta_k))$
    $\lambda \leftarrow \text{prox}_{\eta_2 \epsilon_k g_k^\dagger}(\lambda - \eta_2 z)$
    $u \leftarrow 2\lambda - \lambda_{\text{old}}$
**end for**
Return $\lambda$

---

Starting with $\lambda_2^{(0)} = \cdots = \lambda_K^{(0)} = 0_d$, we propose to apply the following updates at time $t \geq 0$ and so for all $k \in [[2, K]]$:

$$\lambda_k^{(t+1)} = \arg\min_{\lambda_k} \epsilon_1 g_1^\dagger \left( \beta_k^{(t)} - \lambda_k \right) + \epsilon_k g_k^\dagger(\lambda_k) \tag{16}$$

where $\beta_k^{(t)} := \nabla \mathcal{L}(\theta) - \sum_{i \neq k} \lambda_i^{(t)}$. In order to solve (16), we leverage the so-called Chambolle-Pock algorithm Chambolle & Pock (2011). Let us denote $h_1(\lambda) := \varepsilon_1 g_1^\dagger(\beta_k^{(t)} - \lambda)$ and $h_k(\lambda) := \varepsilon_k g_k^\dagger(\lambda)$. Then we can write

$$\inf_\lambda h_1(\lambda) + h_k(\lambda) = \inf_\lambda h_k(\lambda) + \sup_z \langle z, \lambda \rangle - h_1^*(z)$$
$$= \inf_\lambda \sup_z \langle z, \lambda \rangle - h_1^*(z) + h_k(\lambda)$$

where $h_1^*$ is the Fenchel-Legendre transform of $h_1$ given by $h_1^*(x) = \langle x, \beta_k^{(t)} \rangle + \mathbf{1}_{\mathcal{B}_1(\epsilon_1)}(x)$ where $\mathcal{B}_1(\epsilon_1)$ is the ball induced by the norm $g_1$ of radius $\varepsilon_1$. We are now ready to present the Chambolle-Pock algorithm for our setting as presented in Algorithm 5. This algorithm requires to have access to the projection operation w.r.t the norm $g_1$ and the proximal operator w.r.t the norm $g_k^\dagger$, that is, it requires to have access to:

$$\text{proj}_{\mathcal{B}_1(\varepsilon_1)}(x) := \arg\min_{z:\, g_1(z) \leq \varepsilon_1} \|z - x\|_2$$

$$\text{prox}_{\lambda g_k^\dagger}(x) := \arg\min_z \frac{\|z - x\|_2^2}{2} + \lambda g_k^\dagger(z)$$

Computing proximal and projection operators of norms and their duals can also be done using the Moreau decomposition property which states that:

$$\text{prox}_f(x) + \text{prox}_{f^*}(x) = x$$

in particular if $f := \|\cdot\|$ is a norm, we have:

$$\text{prox}_{\|\cdot\|}(x) + \text{proj}_{\mathcal{B}_*(1)}(x) = x$$

where $\mathcal{B}_*(1)$ is the unit ball of the dual norm of $\|\cdot\|$. Finally, the full coordinate gradient scheme is presented in Algorithm 6 which returns a solution of the primal problem defined in (12).

**Lemma C.3.** *Let $\|\cdot\|$ be a norm on $\mathbb{R}^d$ with dual norm $\|x\|_* := max_{z:\|z\| \leq 1}\langle z, x \rangle$, then the Fenchel-Legendre transform of $\|\cdot\|$ is the indicator function of the unit ball induced by its dual norm. More formally, we have*

$$\sup_{z \in \mathbb{R}^d} \langle z, x \rangle - \|z\| = \begin{cases} 0 & \text{if } \|x\|_* \leq 1 \\ +\infty & \text{otherwise} \end{cases}$$

---

**Algorithm 6** Coordinate Gradient Descent to solve (15)

---

**Input:** the gradient $\nabla \mathcal{L}(\theta)$ and $\epsilon_1, \ldots, \epsilon_K > 0$
Initialize $\lambda_2 = \cdots = \lambda_K = 0_d$.
**for** $t = 1$ **to** $T$ **do**
    **for** $k = 2$ **to** $K$ **do**
        $\beta_k^{(t)} \leftarrow \nabla \mathcal{L}(\theta) - \sum\limits_{i \neq k} \lambda_i^{(t)}$

        $\lambda_k^{(t+1)} \leftarrow \arg\min_\lambda h_1(\lambda) + h_k(\lambda)$ with Alg. 5
    **end for**
**end for**
Find $k$ such that $\lambda_k > 0$
Return $x^* := \varepsilon_k \arg\max\limits_{z:\, g_k(z) \leq 1} \langle z, \lambda_k \rangle$

---

*Proof.* Using the fact that $\|x\| = \sup_{z: \|z\|_* \leq 1} \langle z, x \rangle$, we have:

$$
\sup_{z \in \mathbb{R}^d} \langle z, x \rangle - \|z\| = \max_{z \in \mathbb{R}^d} \langle z, x \rangle - \sup_{y: \|y\|_* \leq 1} \langle y, z \rangle
$$
$$
= \sup_{z \in \mathbb{R}^d} \inf_{y: \|y\|_* \leq 1} \langle z, x - y \rangle
$$
$$
= \inf_{y: \|y\|_* \leq 1} \sup_{z \in \mathbb{R}^d} \langle z, x - y \rangle
$$
$$
= \inf_{y: \|y\|_* \leq 1} \begin{cases} 0 \ \text{ if } \ y = x \\ +\infty \ \text{ otherwise} \end{cases}
$$

which gives the desired result. Note that the third equality follows from Sion's minimax theorem. $\square$

**Lemma C.4.** *Let* $\| \cdot \|$ *a norm on* $\mathbb{R}^d$ *and* $\varepsilon > 0$. *Then we have:*

$$
\partial \mathbf{1}_{\mathcal{B}(\varepsilon)}(x) = \begin{cases} \{0\} \ \textit{if } \|x\| < \varepsilon \\ \emptyset \ \textit{if } \|x\| > \varepsilon \\ \{p: \ \langle p, x \rangle = \varepsilon \|p\|_* \} \ \textit{if } \|x\| = \varepsilon \end{cases}
$$

*where* $\| \cdot \|_*$ *is the dual norm of* $\| \cdot \|$, *and* $\mathcal{B}(\varepsilon)$ *is the ball of radius* $\varepsilon$ *w.r.t the norm* $\| \cdot \|$.

*Proof.* Recall that the definition of the subdifferential is:

$$
\partial \mathbf{1}_{\mathcal{B}(\varepsilon)}(x) := \{p: \ \mathbf{1}_{\mathcal{B}(\varepsilon)}(y) \geq \mathbf{1}_{\mathcal{B}(\varepsilon)}(x) + \langle p, y - x \rangle \ \forall y \}
$$

If $\|x\| < \varepsilon$, then we have that $p$ must satisfy for all $y \in \mathcal{B}(\varepsilon)$:

$$
\langle p, y - x \rangle \leq 0
$$

By taking $\gamma$ sufficiently small we can therefore choose $y = x + \gamma \frac{p}{\|p\|_2} \in \mathcal{B}(\varepsilon)$ which leads to

$$
\gamma \|p\|_2 \leq 0
$$

which is only true for $p = 0$ as $\gamma$ can be selected to be negative or positive. Now if $\|x\| > \varepsilon$, then the subdifferential is clearly empty. Finally, let us consider the case where $\|x\| = \varepsilon$. We deduce that:

$$
\langle p, x \rangle \geq \langle p, y \rangle - \mathbf{1}_{\mathcal{B}(\varepsilon)}(y)
$$

and so for all $y$. Therefore we obtain that

$$
\langle p, x \rangle \geq \sup_y \langle p, y \rangle - \mathbf{1}_{\mathcal{B}(\varepsilon)}(y)
$$
$$
= \varepsilon \|p\|_*
$$

But we also have that:

$$
\varepsilon \|p\|_* = \|p\|_* \|x\| \geq \langle p, x \rangle
$$

from which follows that $\langle p, x \rangle = \varepsilon \|p\|_*$ which conclude the proof. $\square$

# D  Further ablations

## D.1  On Adam hyperparameters sweep

In our main results presented in Section 5, the hyperparameters of Adam ($\beta_1$, $\beta_2$, $\epsilon$ etc) follows the setups of (Zhao et al., 2024a). Moreover, weight decay is not used. Below, we further fine-tune those hyperparameters and comapare with our method. We conducted an additional parameter sweep for lr, $\beta_1$, $\beta_2$, weight decay, and $\epsilon$ specifically for the 1B model scale, and obtained the following optimal values:

- Learning rate: 0.0007
- Betas: (0.9, 0.95)
- $\epsilon$: 1e-8
- Weight decay: 0.1 (we observed that a weight decay of 0.1 outperforms no weight decay, though we did not perform an exhaustive search)

We further trained the 1B model on 20B tokens using Adam with these optimal settings and our method. Apart from hyperparameters, the general training setup still follows Zhao et al. (2024a). The following table summarizes the test loss (lower is better) at various training steps:

Table 5: Comparison of the test loss obtained during training when training 1B LLaMA with SinkGD v.s. 1B LLaMA under optimally tuned Adam.

| Method | 40K | 80K | 120K | 150K |
|--------|-----|-----|------|------|
| Adam | 2.880 | 2.728 | 2.659 | 2.651 |
| SinkGD | **2.799** | **2.658** | **2.578** | **2.561** |

For our method, we still use the default parameter setting described in Appendix A. Our method still outperforms Adam by a large margin despite we spent much more compute to sweep the Adam hyperparameters.

