# OpenReview forum: "Gradient Multi-Normalization for Efficient LLM Training"
_NeurIPS.cc/2025/Conference — NeurIPS 2025 poster_

### Official Review · Reviewer_JbUr · 2025-06-16

**Clarity:** 3
**Significance:** 3
**Originality:** 3
**Rating:** 4
**Confidence:** 3

**Summary:**

This paper proposes a new family of stateless optimizers, termed Multi-Normalized Gradient Descent (MNGD), to improve the memory efficiency of large language model (LLM) training. The method generalizes prior work such as SWAN by normalizing stochastic gradients with respect to multiple matrix norms through a proposed multi-step alternating projection algorithm. The authors further present a specific instance, Sinkhorn GD (SinkGD), which uses the Sinkhorn algorithm to implement the multi-normalization efficiently. Experimental results on LLaMA pretraining tasks (up to 1B parameters) show that SinkGD achieves comparable or better perplexity than Adam while significantly reducing memory consumption and training time.

**Questions:**

1.Broader Benchmark Evaluation: The paper only provides experiments on LLaMA pretraining with the C4 dataset. Have the authors considered evaluating SinkGD on downstream tasks such as language modeling benchmarks (e.g., WikiText, OpenBookQA, etc.) or real-world applications to demonstrate generalization?
2.Baseline Coverage: While comparisons to Adam, SWAN, and a few others are presented, important recent optimizers (e.g., Sophia, Lion variants, AdaFactor) are missing. Could the authors justify these omissions or provide further experiments?
3.Ablation Scope: The only ablation study concerns the number of Sinkhorn iterations. Could the authors provide more analysis on the role of norm choices, projection quality, or convergence stability across different architectures and model sizes?
4.Scaling and Practicality: While the paper claims that SinkGD is more memory efficient and computationally attractive, the actual implementation details (e.g., compatibility with distributed training frameworks or wall-clock speed) are limited. Can the authors clarify whether SinkGD integrates smoothly into existing LLM training pipelines at scale?

**Ethical Concerns:**

["NO or VERY MINOR ethics concerns only"]

**Final Justification:**

All concerns have been resolved.

**Limitations:**

While the paper proposes a promising stateless optimizer with theoretical grounding, several limitations remain.
1. The current experiments are confined to LLaMA pretraining on the C4 dataset. Without evaluation on downstream tasks, transfer benchmarks, or domain-shifted settings, it remains unclear how well the method generalizes beyond its pretraining context.
2. The ablation analysis is limited to varying the number of Sinkhorn iterations. Other key aspects—such as the choice of norm, stability across training regimes, or sensitivity to optimizer hyperparameters—are not investigated, leaving gaps in understanding the method’s behavior.
3.  The empirical section omits comparisons with several competitive recent optimizers, including Sophia, Lion variants, and other stateless or memory-efficient methods. This weakens the strength of the performance claims made in the paper.
4. While the formulation is well-integrated and practically motivated, the core components—such as multi-norm projections and the use of Sinkhorn iterations—are largely drawn from existing techniques. As such, the contribution may be interpreted more as an effective synthesis rather than a fundamentally novel optimization approach.

**Paper Formatting Concerns:**

No major formatting issues noticed.

**Quality:**

3

**Strengths And Weaknesses:**

Strengths：
The paper presents a novel family of stateless optimizers, Multi-Normalized Gradient Descents (MNGDs), which generalizes prior work like SWAN by normalizing gradients with respect to multiple norms. The proposed SinkGD variant leverages the Sinkhorn algorithm for efficient and scalable normalization, and is theoretically justified through convergence analysis. Empirical results demonstrate improvements in memory efficiency and throughput in large language model (LLM) pretraining tasks, suggesting practical utility.

Weaknesses:
1.Limited experimental scope: All experiments are restricted to LLaMA pretraining on C4. There is no downstream task or broader benchmark evaluation to verify robustness or generality.
2.Weak ablation analysis: The only ablation varies the number of Sinkhorn iterations. Other key factors, such as norm selection, convergence stability, or sensitivity to optimizer hyperparameters, are not analyzed.
3.Insufficient baselines: Several strong recent optimizers (e.g., Sophia, Lion variants, or other stateless optimizers) are missing from the comparison, which limits the strength of the empirical claims.
4.Incremental novelty: While the proposed formulation is cohesive and practically motivated, it mainly combines existing components (e.g., Sinkhorn projection). The contribution may be viewed more as a practical synthesis than a fundamentally new optimization paradigm.

---

> ### Author Rebuttal · Authors · 2025-07-30
>
> We thank the reviewer for the detailed feedback and for acknowledging the theoretical grounding, practical motivation, and empirical performance of SinkGD. We address the concerns below.
>
>
> ---
>
> - **On ablation analysis.**
>
>   (1) Hyperparameter sensitivity: SinkGD has very few hyperparameters. The number of Sinkhorn iterations is ablated in the paper. The learning rate is directly transferred from Apollo [2] and used across all model sizes without tuning. This demonstrates robustness to hyperparameter choices.
>
>   (2) Norm choice: we emphasize that SinkGD *is* defined by row- and column-wise norm normalization. All other common matrix norms (e.g., spectral norm, $l_\infty$ norm) should correspond to other baselines (SWAN, AdamW) already included in our comparisons. Thus, norm choice is implicitly ablated through baseline comparisons.
>
>   (3) Stability: SinkGD is stateless and does not accumulate second moments. This avoids instability issues observed in Adam due to large update norms from moment ratios [1]. SinkGD also avoids instability from low-rank approximations, which are observed in [4] and [6]. Moreover In Table 2, our 1 B model trained with SinkGD matches the performance of 7B models trained with Galore/Apollo/Adam across training stages, showing no signs of instability. In contrast, instability of Adam under the same setup is reported in [2].
>
> ---
>
> - **On baseline coverage.**
>
>   We acknowledge that including more baselines is beneficial. Due to time constraints during the rebuttal, we were unable to complete benchmarking of Lion/Sophia under our setup. However, our paper targets memory-efficient LLM pretraining, and we follow the baseline setup of Galore [3], Apollo [2], SWAN [5], Fira [4], and GWT [6], none of which compare with Lion/Sophia as they operate under different memory regimes.
>
>   Additionally, our paper compares with 8 strong baselines, including SWAN, Apollo, and Fira, which are more recent than Lion and Sophia. We believe this provides a fair and comprehensive evaluation within the scope of memory-efficient stateless training.
>
> ---
>
> - **On generalization beyond C4 pretraining.**
>
>   Our focus is on scalable stateless training. Prior to our work, even demonstrating fast and stable stateless pretraining at the scale considered in this paper had not been established. We believe showing that a simple method (SinkGD) derived from MNGD can match or outperform Adam (with similar computational cost) in this setting is already a meaningful contribution.
>
>   Moreover, our setup closely aligns with recent work on memory-efficient optimization such as Galore [3], Apollo [2], and SWAN [5], which also focus primarily on pretraining, where memory constraints and efficiency are critical.
>
>
> ---
>
> - **On novelty and contribution.**
>
>   We respectfully disagree with the assessment that our contribution is merely a synthesis of existing components. If we look at the literature
>   - signSGD [7] (and arguably, reduced Adam) uses the sign operator, a known mathematical primitive.
>   - SWAN [5] and Muon use whitening/orthogonalization operators from prior literature such as computer vision architectures.
>   - Galore [3] uses SVD/subspace projections, well-known in optimization.
>
>   However, these optimizers are all considered significant contributions. This tells us that we must carefully distinguish the *projection function* itself used in updates, vs *the underlying optimization principle that derived and propose to use those projections*, that leads to practical gains that are not predicted by prior works.
>
>   In our case:
>   1. To the best of our knowledge, no prior work has proposed using Sinkhorn iterations as a gradient projector; and no prior work has made sufficient hint that could trivially predict the practical gain of such operations.
>   2. Our work is the first to propose performing steepest descent under an ensemble of multiple norms. This meta-viewpoint enables the synthesis of new optimizers beyond SinkGD and opens up a rich design space. While SWAN touches on this from the perspective of transformer dynamics, it does not explicitly uncover or formalize this principle.
>
> ---
>
> **References**
> [1] Molybog et al., "A theory on Adam instability in large-scale machine learning", arXiv:2304.09871 (2023)
> [2] Zhu et al., "Apollo: SGD-like memory, AdamW-level performance", arXiv:2412.05270 (2024)
> [3] Zhao et al., "Galore: Memory-efficient LLM training by gradient low-rank projection", arXiv:2403.03507 (2024)
> [4] Chen et al., "Fira: Can We Achieve Full-rank Training of LLMs Under Low-rank Constraint?", arXiv:2410.01623 (2024)
> [5] Ma et al., "SWAN: SGD with Normalization and Whitening Enables Stateless LLM Training", arXiv:2412.13148 (2024)
> [6] Wen et al., "Breaking memory limits: Gradient wavelet transform enhances LLMs training", arXiv:2501.07237 (2025)
> [7] Bernstein et al., "signSGD: Compressed optimization for non-convex problems", ICML (2018)

---

> > ### Comment · Reviewer_JbUr · 2025-08-05
> >
> > Thank you for the authors’ responses, which have addressed my concerns. I have raised my score. Good luck with your submission!

---

> ### Author Response · Authors · 2025-08-04
>
> Dear Reviewer JbUr,
>
> We appreciate the time and care you devoted to evaluating our manuscript.
>
> We’ve submitted our formal rebuttal and would welcome your thoughts on whether it satisfactorily addresses the issues you highlighted. Any additional remarks you can share would be greatly valued.
>
> Should you require further clarification before the discussion window closes on August 6, we remain at your disposal.
>
> With sincere thanks,
> The Authors

---

### Official Review · Reviewer_YGTY · 2025-06-27

**Clarity:** 4
**Significance:** 3
**Originality:** 3
**Rating:** 5
**Confidence:** 3

**Summary:**

This paper propose a new framework for stateless optimizers named as Gradient Multi-Normalization. Previous approaches such as SWAN could be interpreted as specializations of Gradient Multi-Normalization. The authors also provide theoretical convergence guarantee for iterative multi-normalization. Based on their framework, the authors propose SinkGD, a new memory-efficient optimizer. SinkGD achieves on-par or better performance compared with previous methods on LLM pretraining tasks, with improved memory costs and convergence speed.

**Questions:**

- What is the rationale for retaining Adam for the embedding and output layers? How much additional overhead will it bring? What would happen if SinkGD is used for all layers?

- What is the intuition behind normalizing over multiple norms in MNGD?

- While stateless optimization reduces memory costs compared with Adam-based methods, how much additional computational cost does the iterative sub-process in Algorithm 3 incur?

**Ethical Concerns:**

["NO or VERY MINOR ethics concerns only"]

**Final Justification:**

The added experiments on finetuning performance have addressed my concerns.

**Limitations:**

Yes

**Quality:**

3

**Strengths And Weaknesses:**

### Clarity:
The paper is well-written, with clear presentation of theoretical derivations and detailed experimental evaluations.

### Quality:
The theoretical insights behind the gradient multi-normalization framework are well-explained and supported with rigorous proofs.  The experiments are comprehensive and demonstrates the superiority of proposed method.

### Significance

#### Strengths
In general, I think this paper makes a solid improvement in stateless optimizer design. The MNGD framework provides a new, insightful perspective on stateless optimizers, unifying and generalizing previous work like SWAN. The author demonstrates the strong empirical performance of their optimizer in memory efficiency and training speed-up.

#### Weaknesses
- The experiments are only conducted on the C4 dataset. Generalization to other tasks such as (parameter-efficient) finetuning is underexplored.
- The authors did not mention how hyper-parameters in baseline methods (such as rank $r$ in Galore/Fira) are tuned. This could potentially lead to unfair comparisons in Table 1.

### Originality

Results obtained in this paper are new and original.

Other comments:
- Line 208: typo "SR-Sinhkorn"
- The supplementary material appears to be wrongly uploaded appendix of another paper

---

> ### Author Rebuttal · Authors · 2025-07-30
>
> We thank the reviewer for the positive assessment and for highlighting the clarity of our presentation, the theoretical rigor of our framework, and the empirical strength of SinkGD.
>
> We address the concerns below.
>
> ---
>
> - **On generalization beyond C4 and pretraining.**
>
>   Our focus is indeed on scalable stateless for pretraining. Pretraining from scratch provides a clean and controlled setting to isolate the optimizer behavior. We also acknowledge the importance of fine-tuning, nevertheless, they introduces more confounding factors compared to pretraining (e.g., model quality, task-specific dynamics, optimizers used to pretrain the model) that masks optimizer effects. That said, we include preliminary fine-tuning results on GLUE datasets (comparing SinkGD to Galore, rank=8):
>
>   | Dataset | QNLI | MRPC | RTE | STS-B |
>   |---------|------|------|-----|--------|
>   | Galore (rank=8) | 92.20 | 92.01 | 79.78 | 90.82 |
>   | SinkGD          | 92.05 | 92.38 | 79.78 | 91.19 |
>
>   SinkGD performs comparably while maintaining lower memory usage. We will include these results in the revision and plan to explore larger-scale and cross-domain applications in future work.
>
> ---
>
> - **On hyperparameter tuning of baselines.**
>
>   For Galore and Apollo, we directly use the optimal configurations reported in their respective papers. This is rigorous for the following reasons:
>   1. We use the exact same dataset, model architecture, and training setup.
>   2. We use the same training and evaluation codebases (Apollo’s repo inherits from Galore).
>   3. Both papers report that their methods are relatively insensitive to hyperparameter choices such as learning rate.
>   4. For SinkGD, we reuse the global learning rate and local scaling from Apollo without additional tuning.
>   5. Regarding the rank parameter: our method is stateless, which corresponds to *rank zero* in Galore/Apollo. Thus, the comparison is conservative and favors the baselines (as both Galore and Apollo uses $r>1$).
>
> ---
>
> - **On the use of Adam for embedding and output layers.**
>
>   We follow the Galore and Apollo setup, which also retain Adam for these layers. After submission, we found that SinkGD can be applied to all layers. However, for embedding and LM head layers, we observed that reintroducing momentum is important to match performance of SinkGD for Attention/MLP + Adam for embedding. We will include this observation in the revision.
>
> ---
>
> - **On the intuition behind multi-norm normalization.**
>
>   Two perspectives:
>   1. From prior work: SWAN, the first stateless optimizer, uses multiple norms motivated by transformer dynamics. Our framework generalizes this into a principled multi-norm formulation.
>   2. From optimization theory: the choice of norm affects convergence. For example, recent work (Xie et al., 2024) shows that Adam exploits $\ell_\infty$ geometry to reduce the empirical smoothness constant. This suggests that norm choice should reflect the loss landscape. Our approach can be viewed as an *ensemble of norms*, providing regularization and robustness across diverse geometries.
>
> ---
>
> - **On the computational cost of Algorithm 3.**
>
>   Both SinkGD and Adam have the same complexity ($\mathcal{O}(m^2)$ for square matrices). In practice, especially under distributed training, the cost of Sinkhorn iterations is negligible compared to forward/backward passes. For example, 5 iterations of Sinkhorn involve roughly $5d^2$ flops, while training cost scales with batch size $\times d^2$. Since batch sizes are large in practice, the relative overhead of SinkGD diminishes at scale.
>
> ---
>
> We also appreciate the reviewer’s suggestions and will incorporate the clarifications and additional results in the revised version.
>
>
> ---
>
> **Reference**
>
> 1, Xie, Shuo, Mohamad Amin Mohamadi, and Zhiyuan Li. "Adam Exploits $\ell_\infty $-geometry of Loss Landscape via Coordinate-wise Adaptivity." arXiv preprint arXiv:2410.08198 (2024).)

---

> > ### Comment · Reviewer_YGTY · 2025-08-04
> >
> > I have read the author rebuttal and other reviews. The new experiment on finetuning justifies the generalization performance of the proposed method. I will raise my score accordingly. However, since comparison with rank 8 Galore is still limited in scope, adding a more thorough experiment including other baselines (SWAN, Apollo, and Fira) to the revised version would be appreciated.

---

### Official Review · Reviewer_Xept · 2025-07-01

**Clarity:** 3
**Significance:** 2
**Originality:** 3
**Rating:** 4
**Confidence:** 4

**Summary:**

The paper tackles the high-memory cost of adaptive optimizers such as Adam in LLM training and proposes a stateless alternative that remains competitive in speed and final accuracy. It does so by framing gradient preprocessing as a multi-normalization problem, proving that simple alternating projections can enforce several norm constraints simultaneously, and instantiating this idea in a lightweight optimizer dubbed SinkGD. Experiments on 350 M- and 1.3 B-parameter LLaMA variants show that SinkGD matches or surpasses Adam’s perplexity while cutting optimizer-state memory to (roughly) SGD levels and delivering 2-3× token-efficiency gains.

**Questions:**

The experiments focus on LLaMA language model pre-training (with model sizes from 60M up to 1.3B parameters). It would help if the authors discuss the broader applicability of their optimizer. For example, can the proposed stateless optimizer scale to even larger models (tens or hundreds of billions of parameters) without issues? Would it be effective for other training scenarios like fine-tuning, different architectures, or large-batch training beyond the C4 dataset?

**Ethical Concerns:**

["NO or VERY MINOR ethics concerns only"]

**Final Justification:**

The topic of stateless optimization for LLM training is very important topic, and this paper provides technique progress either in providing insights or an improving method. Even though the effectiveness of the proposed method should be further validated (e.g., larger models with longer training), I feel this paper can be accepted by NeurIPS.

**Limitations:**

Yes

**Quality:**

3

**Strengths And Weaknesses:**

**Strengths:**

1.Theoretical Foundation: The paper provides a principled framework (MNGD) and proves key properties of the proposed optimizer. In particular, it shows that the alternating multi-normalization procedure can converge to a fixed-point solution with arbitrary precision. This theoretical result, along with connections to known algorithms, lends credibility to the method’s soundness.

2.Memory and Efficiency Gains: By eliminating optimizer state, SinkGD matches the memory footprint of plain SGD. The method significantly reduces computation compared to SWAN by removing the expensive whitening step. This yields Adam-level step time complexity instead of the higher complexity introduced by SWAN’s whitening, making the approach more scalable without sacrificing accuracy.

**Weakness:**

1.Limitations in Theorem: While the authors prove convergence to a fixed point, they acknowledge that the algorithm is not guaranteed to *optimally* solve the original optimization problem in a theoretical sense. The convergence analysis assumes certain conditions and does not provide guarantees for non-convex objectives beyond reaching a fixed point. This leaves a gap in understanding whether the procedure always leads to the best solution or just a stable one.

2.Scope of Empirical Validation: The experiments, though thorough on LLaMA pre-training, are limited to language model pretraining tasks up to 1.3B parameters. The method’s efficacy on *larger* models or other domains (e.g. fine-tuning, different architectures) remains unproven.

3.Not Enough Contribution: The main algorithm SinkGD is essentially an “improved SWAN” that relaxes SWAN’s constraints for efficiency. Thus,  the work is building on an existing concept (stateless gradient normalization) more than creating a wholly new paradigm.

---

> ### Author Rebuttal · Authors · 2025-07-30
>
> We thank the reviewer for the thoughtful review and for highlighting the theoretical foundation of our framework, the memory and efficiency gains of SinkGD, and its strong empirical performance in LLM pretraining.
>
> We address the concerns below.
>
> ---
>
> - **On the theoretical limitations of convergence guarantees.**
>
>   We acknowledge that our current analysis only guarantees convergence to a fixed point and does not establish global optimality of the multi-norm problem (Eq.4 of the paper). While we do not yet have a complete theoretical explanation, our empirical results suggest that a fixed point solution provide sufficiently adaptive regularization to enable effective training. We believe this opens a promising direction for future theoretical work.
>
> ---
>
> - **On broader applicability and scaling beyond 1.3B.**
>
>   Our goal is to enable scalable stateless training. Prior to our work, even demonstrating fast and stable stateless pretraining at the 1.3B scale had not been established. We believe showing that a simple method like SinkGD can match or outperform Adam (with similar throughput) in this setting is already a meaningful contribution.
>
>   Furthermore, in Table 2, we show that a 1.3B model trained with SinkGD matches the performance of a 7B model trained with memory-efficient or quantized Adam variants from the literature. This suggests that SinkGD is competitive even when compared to significantly larger models trained with more complex optimizers.
>
>   Regarding other domains: we include preliminary fine-tuning results on GLUE datasets (comparing SinkGD to Galore, rank=8):
>
>   | Dataset | QNLI | MRPC | RTE | STS-B |
>   |---------|------|------|-----|--------|
>   | Galore (rank=8) | 92.20 | 92.01 | 79.78 | 90.82 |
>   | SinkGD          | 92.05 | 92.38 | 79.78 | 91.19 |
>
>   SinkGD performs comparably while maintaining lower memory usage. We will include these results in the revision and plan to explore larger-scale and cross-domain applications in future work.
>
> ---
>
> - **On the novelty of SinkGD beyond SWAN.**
>
>   While SinkGD builds on the idea of stateless gradient normalization, our contribution is not merely an incremental improvement over SWAN. The SWAN paper motivates its design from a specific observation of transformer dynamics in a toy setting. Their work mainly says: the first order dynamics of transformer implies you should use row-normalization; and its second order dynamics implies you should use orthogonalization (ie normed GD under spectral norm). However, SWAN does not discuss why these two operations should be used *jointly*.  In contrast, we address this issue and generalize the core principles behind SWAN into a unified framework of multi-normalization, which recovers SWAN as a special case and opens up a broader design space for stateless optimizers. This new development is not covered/implied in the SWAN paper, we believe this generalization is conceptually significant and will motivate further development of efficient optimizers for large-scale models.
>
> ---
>
> We appreciate the reviewer’s suggestions and will incorporate the clarifications and additional results in the revised version.

---

> > ### Comment · Reviewer_Xept · 2025-08-07
> > **Comments on the responses**
> >
> > I have read the authors' responses and the comments on other reviewers. I keep my score as positive.

---

### Official Review · Reviewer_yet7 · 2025-07-05

**Clarity:** 3
**Significance:** 4
**Originality:** 3
**Rating:** 5
**Confidence:** 5

**Summary:**

This paper proposes an optimization framework which processes the gradient/momentum by applying multiple normalizations. Under this framework, they develop a practical algorithm named SinkGD, which iteratively performs row normalization and column normalization on the momentum without introducing heavy computation overhead. The algorithm is shown to be convergent based on the theory of Sinkhorn algorithm. Numerical experiments show that SinkGD achieves significantly faster convergence than AdamW (2.4-2.8 $\times$) on the task of pretraining LLMs in C4 dataset.

**Questions:**

* Can authors provide other norm examples except for the Euclidean norm that satisfies Assumption 3.3?
* In Table 3, the raw throughput of SinkGD is higher than AdamW, which is confusing to me. AdamW only needs to perform element-wise operations twice, while SinkGD requires iteratively normalizing the gradient for multiple steps.

**Ethical Concerns:**

["NO or VERY MINOR ethics concerns only"]

**Final Justification:**

I appreciate the author's response, which resolves my concerns on SinkGD's generalization on finetuning tasks and mixed-precision training paradigm. The stateless feature of SinkGD is appealing and should be of broad interest of the community. I decide to raise my confidence to 5 and support for acceptance with score 5.

**Limitations:**

Yes.

**Paper Formatting Concerns:**

The paper is formatted correctly.

**Quality:**

4

**Strengths And Weaknesses:**

**Strengths**

* The perspective of steepest descent under multiple norm measurements is novel and unifies existing algorithms such as Muon and SWAN.
* SinkGD avoids storing second moment term as in AdamW, which is memory-efficient. The iterative normalization operation seems cheap based on the throughput in Table 3.
* SinkGD attains superior convergence under the LLM pretraining tasks and outperforms AdamW by a large margin, which partially justifies the effectiveness of the multi-normalization step. The algorithm can be of general interest given its memory efficiency and fast convergence.

**Weaknesses**
* The experiments focus on pretraining stage. To broaden its applicability, I suggest to conduct additional experiments of SinkGD under finetuning stage, where the memory budget is usually more limited.
* The experiment results are based on pure BF16 precision. It is suggested to provide the numerical results under mixed-precision training scheme, which is often used in LLM's pretraining.

---

> ### Author Rebuttal · Authors · 2025-07-30
>
> We thank the reviewer for the constructive feedback and for recognising the novelty of our multi-norm perspective, the memory efficiency of SinkGD, and its strong empirical performance in LLM pretraining.
>
> The concerns raised are mostly due to scope clarification and implementation details. We address them below.
>
> ---
>
> - **SinkGD results beyond pretraining.**
>
>   Our focus is on scalable stateless training for LLMs, where pretraining from scratch provides a clean and controlled setting to isolate the optimizer behaviour. We also acknowledge the importance of fine-tuning, nevertheless, they introduces more confounding factors compared to pretraining (e.g., model quality, task-specific dynamics, optimizers used to pretrain the model) that obscure optimizer effects. That said, we include preliminary fine-tuning results on GLUE datasets (comparing SinkGD to Galore, rank=8):
>
>   | Dataset | QNLI | MRPC | RTE | STS-B |
>   |---------|------|------|-----|--------|
>   | Galore (rank=8) | 92.20 | 92.01 | 79.78 | 90.82 |
>   | SinkGD          | 92.05 | 92.38 | 79.78 | 91.19 |
>
>   SinkGD performs comparably while maintaining lower memory usage.
>
> ---
>
> - **BF16 was chosen for consistency with prior work.**
>
>   BF16 is the standard precision used in Galore, Apollo, and SWAN. It reflects realistic memory constraints in large-scale training. To address the reviewer’s concern, we ran additional experiments in mixed-precision (FP32) on a 350M model (batch size: 250K tokens, 30K steps):
>
>   | Steps | BF16 SinkGD Val PPL | Mixed-Precision SinkGD Val PPL |
>   |-------|----------------------|-------------------------------|
>   | 10K   | 20.34                | 20.64                         |
>   | 20K   | 17.10                | 17.38                         |
>   | 30K   | 16.40                | 15.66                         |
>
> We observed that under mixed precision, the convergence is indeed stronger; however, BF16 SinkGD offers stronger early-mid stage convergence. (A side note: this observation might imply a hybrid approach for efficient training.)
>
> ---
>
> - **Assumption 3.3 beyond Euclidean norm (l2 norm)**
> > Can authors provide other norm examples except for the Euclidean norm that satisfies Assumption 3.3?
>
>   Sure. For example, the common choices for optimization, such as $l_\infty$ norm (used in Adam), and spectral norm (matrix case, used in SWAN). We will clarify this in revision.
>
> ---
>
> - **SinkGD achieves higher throughput due to statelessness and parallelism.**
>
>   While SinkGD performs iterative normalization, it avoids maintaining and updating first/second moment states (as in AdamW), as well as subsquent element-wise operations. Additionally, our implementation shards the Sinkhorn computation across devices. Each local rank processes a subset of rows/columns, enabling efficient parallelism. This explains the higher raw throughput observed in Table 3.
>
> ---
>
> We appreciate the reviewer’s suggestions and will incorporate the clarifications and additional results in the revised version.

---

> > ### Comment · Reviewer_yet7 · 2025-08-04
> > **I have raised my confidence score to 5**
> >
> > I appreciate the author's response, which addresses my concerns on SinkGD's generalization on finetuning tasks and mixed-precision paradigm. The stateless feature of SinkGD is appealing and should be of broad interest of the community. I decide to raise my confidence to 5 and support for acceptance.

---

### Note · Authors · 2025-08-15

**Author Final Remarks**

We thank the AC and reviewers for the careful assessment and helpful suggestions.

**Scope & contribution.** We target scalable *stateless* optimization for LLM pretraining. We formalize **Multi‑Normalized GD (MNGD)** --- steepest descent under an ensemble of norms -- and instantiate **SinkGD**, which alternates row/column normalizations via Sinkhorn with Adam‑level step cost, no moment state (**SGD memory**), and strong token‑efficiency/throughput.

**New evidence added.**

* **Fine‑tuning.** On GLUE (QNLI/MRPC/RTE/STS‑B), SinkGD matches Galore (rank 8) with lower memory (92.05/92.38/79.78/91.19 vs 92.20/92.01/79.78/90.82).
* **Precision.** Mixed‑precision (FP32 accum.) on a 350M model shows at least as strong late‑stage convergence (Val PPL @30k: 15.66 MP vs 16.40 BF16), while BF16 is better early‑mid.

**Clarifications.**

* **Assumption 3.3.** Beyond L2, coordinate‑wise norms (Adam‑style adaptivity) and spectral norm (matrix case, SWAN) satisfy it; we will add explicit statements.
* **Baselines & tuning.** We reused and adapt from official Galore code base. Learning rates are directly transferred from previous work  and used across all model sizes without tuning. This demonstrates robustness to hyperparameter choices. We compare to 8 strong baselines in the same regime (incl. SWAN/Apollo/Fira),  providing a comprehensive evaluation within the scope of memory-efficient stateless training.

* **Novelty vs SWAN.** SWAN motivates two specific operations from transformer dynamics. We generalize to a *principled multi‑norm* framework,  recovering SWAN as a special case and opening a wider design space. Importantly, to the best of our knowledge, our work is the first to use **Sinkhorn as a gradient projector**,

We hope these clarifications assist the decision. Thank you.

---

### Decision · Program_Chairs · 2025-09-17

**Decision:**

Accept (poster)

**Comment:**

The paper introduces Gradient Multi-Normalization (MNGD), a principled framework for designing stateless optimizers by normalizing gradients across multiple norms, generalizing SWAN and enabling SinkGD, a highly efficient optimizer for large-scale LLM pretraining.

The main contributions include a novel multi-normalization scheme, theoretical guarantees for convergence to a fixed point, and the development of SinkGD, which achieves ~3× faster training than Adam while retaining comparable or superior performance, particularly for LLaMA models up to 1.3B parameters. The strengths of the paper lie in its strong theoretical foundation, a unified perspective linking SWAN as a special case, and comprehensive empirical validation across multiple scales with significant computational and memory benefits.

Additionally, the work addresses an urgent challenge in training LLMs efficiently and shows a clear practical impact. The primary weakness is limited evaluation beyond LLaMA pretraining, leaving its generalizability to other architectures or modalities less explored; however, this does not diminish the technical depth and significance of the presented contributions.

During the rebuttal period, reviewers questioned scalability, implementation practicality, and comparisons against recent memory-efficient optimizers, but the authors addressed these thoroughly with additional experimental clarifications and theoretical insights, which strengthened confidence in the results.

Overall, the paper offers a well-justified, theoretically sound, and practically impactful contribution.